# Noise-Guided Transport: Imitation Learning from Random Priors

**Lionel Blondé** [1]   **Joao A. Candido Ramos** [2][1]   **Alexandros Kalousis** [1]

## Abstract

We consider imitation learning in the low-data regime, where only a limited number of expert demonstrations are available. In this setting, methods that rely on large-scale pretraining or high-capacity architectures can be difficult to apply, and efficiency with respect to demonstration data becomes critical. We introduce Noise-Guided Transport (NGT), a lightweight off-policy method that casts imitation as an optimal transport problem solved via adversarial training. NGT requires no pretraining or specialized architectures, incorporates uncertainty estimation by design, and is easy to implement and tune. Despite its simplicity, NGT achieves strong performance on challenging continuous control tasks, including high-dimensional Humanoid tasks, under ultra-low data regimes with as few as 20 transitions.

## 1. Introduction

The recent advent of pretrained, internet-scale vision–language models has made supervised learning techniques like behavioral cloning (BC) (Pomerleau, 1989) viable for imitation learning (IL) in the very large data regime (Black et al., 2024; Amin et al., 2025), where tens of thousands of state-actions pairs are available. Yet, **in low-data regimes** where only a handful of expert demonstrations are available, BC often fails to generalize. This challenge is common in healthcare applications such as human gait analysis from impaired patients, where acquiring diverse, high-quality demonstrations is constrained by (i) patient availability and (ii) clinical variability. Limited diversity in demonstrations often leads BC to accumulate compounding errors at test time (Ross & Bagnell, 2010). In this work, we devise a sample-efficient method for IL to address scenarios with scarce expert data.

Apprenticeship learning (Abbeel & Ng, 2004)—inverse RL (IRL) in an inner loop; RL in an outer loop—mitigates the compounding errors BC suffers from by being online. Yet, IRL is tedious because it is ambiguous. This ambiguity was identified by Ziebart et al. (2008) and solved with maximum entropy IRL (MaxEnt IRL). Finn et al. (2016) later showed that GANs (Goodfellow et al., 2014) solve the same objective as MaxEnt IRL. These observations align with the sustained success of GAIL (Ho & Ermon, 2016) in IL. Given our focus on sample efficiency—in terms of expert dataset size but also interactions with the world—we turn to the off-policy evolution of GAIL, specifically DAC (Kostrikov et al., 2019) and SAM (Blondé & Kalousis, 2019). Inheriting from GANs, these adversarial IL (AIL) approaches minimize the JS-divergence between the expert and the agent by using a binary classifier that is trained to discriminate between their respective state-action distributions. AIL has then quickly been extended to other divergences and distances (Ghasemipour et al., 2019; Ke et al., 2019).

Our core desideratum is: learning a reward function that distinguishes expert from agent. In this work, we learn a reward with an objective that stems from the problem of learning from random priors, whose foundations we lay out in SECTION 2. In SECTION 4, we derive our reward learning objective, before showing that it coincides with an earth-mover distance, a metric grounded in optimal transport (OT) theory (Villani, 2009). We refer to our method as **Noise-Guided Transport (NGT)**, due to its reliance on guidance from random priors (akin to noise; see SECTION 2) and its OT equivalence. We also provide guarantees for the loss optimized in practice, showing how its deviation from the true objective concentrates with sample size. In SECTION 5, we empirically evaluate and compare NGT in the low-data regime against a diverse set of baselines, including OT-based methods and a diffusion-based AIL method (Wang et al., 2023). We benchmark the methods on standard continuous control tasks for low-data imitation learning. Notably, we include humanoid locomotion: a high-dimensional, challenging control task that is rarely addressed in this regime due to its complex dynamics and large state-action space. We also tackle the state-only setting, where expert actions are not available Among the baselines, only a diffusion-based approach is able to make progress on this task, al-

---

[1]University of Applied Sciences Western Switzerland, Geneva [2]University of Geneva, Switzerland. Correspondence to: Lionel Blondé <mail@lionelblonde.com>.

*Proceedings of the $43^{rd}$ International Conference on Machine Learning*, Seoul, South Korea. PMLR 306, 2026. Copyright 2026 by the author(s).

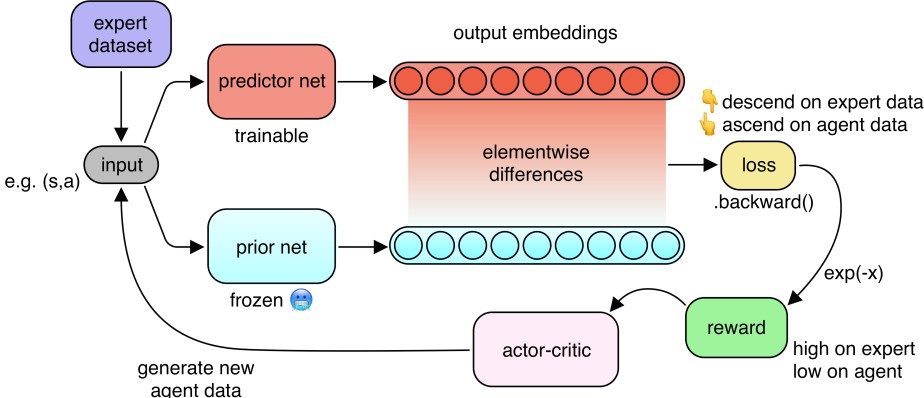

*Figure 1.* Noise-Guided Transport; emphasis on how the reward surrogate is learned.

beit sub-optimally and with greater computational overhead. Overall, our results demonstrate that NGT scales gracefully with both task complexity and data scarcity, all while remaining lightweight.

## 2. Background and Setting

**Learning from demonstrations.** We consider an agent that interacts with a Markov Decision Process (MDP) $(\mathbb{S}, \mathbb{A}, P, r, \gamma)$. $\mathbb{S}$ and $\mathbb{A}$ denote the state and action spaces, $P$ the transition dynamics, $r$ the reward function, and $\gamma \in [0, 1)$ the discount factor. We work in the episodic setting, where $\gamma$ resets to $0$ upon episode termination, reflecting finite-horizon rollouts up to $T$. A policy $\pi(a|s)$ specifies a distribution over actions conditioned on states. The agent acts in the environment by following its policy in order to maximize expected cumulative rewards. In IL, the reward function is *unknown*. The objective is therefore to learn a policy that reproduces the behavior of an expert from a demonstration dataset $\mathcal{E}$, which typically contains trajectories $(s_t, a_t)_{t=1}^{T}$. We want our imitator agent to learn a robust reward signal $r_\xi$ from $\mathcal{E}$. When the demonstrations contain only states, the task becomes to learn a policy whose state occupancy matches that of the expert. Formally, the input space of the reward model $r_\xi$ can be $\mathbb{S} \times \mathbb{A}$, $\mathbb{S} \times \mathbb{S}$ (*state-state* case) or $\mathbb{S}$ (*state-only* case). We will use $\mathbb{X}$ to denote any of these options. $P_{\text{expert}}$ denotes the probability distribution of expert data over $\mathbb{X}$, and $P_{\text{agent}}$ the one described by the agent. Finally, $P(\mathbb{X})$ denotes an arbitrary distribution over $\mathbb{X}$.

**Learning architecture and algorithm.** The reward function is learned jointly with the agent, itself learned via an actor-critic architecture (Crites & Barto, 1995), which comprises the policy (actor) $\pi_\theta$ and a action-value (critic) $Q_\omega : \mathcal{S} \times \mathcal{A} \to \mathbb{R}$. The critic is responsible for assigning credit (from $r_\xi$) to actions (from $\pi_\theta$) in time, and is learned jointly with the policy by generalized policy improvement

(Sutton & Barto, 2018). The three functions are therefore learned jointly, and their updates are interlaced. They are modeled with neural networks, with parameters $\theta$, $\omega$, and $\xi$. In addition, all three are learned in an off-policy fashion. This trait is supported by a replay buffer, enabling the agent to replay past experiences (Lin, 1992). Specifically, NGT learns $\pi_\theta$ and $Q_\omega$ with the Soft Actor-Critic (SAC) off-policy RL algorithm (Haarnoja et al., 2018), with the apprenticeship reward $r_\xi$ learned in an interweaved inner loop. The crux of this work resides in the design of a novel reward learning method—based on the problem of learning from random priors—which we introduce in SECTION 4.

**Learning from random priors.** The problem of *learning from random priors* refers to a prediction setup in which a frozen, randomly initialized neural network provides a deterministic target mapping, and a second network is trained to match its outputs. Let $f_\xi^\dagger : \mathbb{X} \to \mathbb{R}^m$ denote a neural network that is randomly initialized once and frozen thereafter. A predictor network $f_\xi$ identical in architecture but trainable, is optimized to match the outputs of this prior. Given a nonnegative matching loss $\ell$, the problem is: $(\mathfrak{P}) : \inf_\xi \; \mathbb{E}_{x \sim P(\mathbb{X})} \left[ \ell \big( f_\xi(x), f_\xi^\dagger(x) \big) \right]$ where $P(\mathbb{X})$ is an arbitrary distribution over the input space. Because $f_\xi^\dagger$ is fixed, gradient steps on $(\mathfrak{P})$ decrease the discrepancy $\ell \big( f_\xi(x), f_\xi^\dagger(x) \big)$ on samples drawn from $P$, while leaving its behavior largely unconstrained outside the support of $P$. This makes the predictor-prior discrepancy a form of pseudo-density estimator: low values indicate regions where the predictor has adapted to the distribution, and high values indicate regions it has not. In SECTION 4, we derive our own adversarial reward learning objective from this principle.

## 3. Related Works

**OT in AIL.** DAC and SAM (Kostrikov et al., 2019; Blondé & Kalousis, 2019) have gained recognition for their

sample efficiency and simplicity. These methods frame IL as minimizing a divergence between the agent and expert distributions, in line with the original GAN formulation (Goodfellow et al., 2014). This divergence minimization can be interpreted through the lens of OT, both in its primal and dual formulations (Chang et al., 2023). PWIL (Dadashi et al., 2021) approximates the primal form of the EMD with an iterative procedure, while methods like ROT (Haldar et al., 2022) and MAAD (Ramos et al., 2024) use the Sinkhorn algorithm—solving an entropy-regularized approximation of the EMD in the primal formulation—to match full trajectories. In contrast, the dual formulation approximates OT by learning the Kantorovich potentials using neural networks, as done in the adversarial training setup of WGAN (Arjovsky et al., 2017; Peng et al., 2021). NGT also adopts a dual OT perspective. In contrast, AIL-Boost (Chang et al., 2024) adopts an orthogonal strategy by enhancing DAC through boosting, a classic ensemble learning technique.

**Learning from random priors.**  This principle has appeared in several contexts. Early work on randomized value functions showed that injecting fixed random components into value estimates can act as an implicit Bayesian prior and improve generalization (Osband et al., 2014). Randomized prior functions formalized this idea by adding a fixed random network to the value function and interpreting the residual as the learnable component (Osband et al., 2018). A similar mechanism underlies prediction-based pseudo-density estimation (PDE) methods such as Random Network Distillation (RND), where the predictor is trained to match a frozen random target and the prediction error captures distributional mismatch as novelty for exploration (Burda et al., 2018). Such PDE from random priors has also been used to learn an *expert* detector that guides imitation in offline IL, in an approach called Random Expert Distillation (RED) (Wang et al., 2019). Alternatively, such PDE could also be geared towards out-of-distribution (OOD) avoidance, also called anti-exploration, in offline RL. Notably, Rezaeifar et al. (2022) reports that RND is ineffective for anti-exploration in continuous control tasks in offline RL. However, later findings showed that it performs well when the predictor and prior networks use specific asymmetric architectures that learn separate representations for states and actions before merging them (Nikulin et al., 2023). In this work, we demonstrate that NGT can leverage random priors for continuous control beyond feature engineering. In addition, Ciosek et al. (2020) provides theoretical grounding for RND, analyzing concentration properties that characterize the conditions under which the difference between the prior and predictor vanishes.

We expand on related works further in APPENDIX D.

## 4. Method and Guarantees

### 4.1. Reward Learning: Objective

We now introduce our reward learning objective, deriving an adversarial training objective from the problem of learning from random priors introduced in SECTION 2. We also illustrate it in FIGURE 1.

By comparing the outputs of predictor $f_\xi$ and prior $f_\xi^\dagger$ through a non-negative loss $\ell$—formalized in the problem formulation $(\mathfrak{P})$ : $\inf_\xi \ \mathbb{E}_{x \sim P(\mathbb{X})}\left[\ell\big(f_\xi(x), f_\xi^\dagger(x)\big)\right]$—we have at our disposal a task whose complexity can scale depending jointly on several factors: the architecture of the networks $f_\xi$ and $f_\xi^\dagger$, the size of the output embedding $m$, and the properties of the loss $\ell$. In particular, the magnitude and variability of the matching loss carry information about the epistemic uncertainty associated with approximating the random target in $\mathbb{R}^m$. Performing gradient descent on $(\mathfrak{P})$ lowers the discrepancy $\ell\big(f_\xi(x), f_\xi^\dagger(x)\big)$ on samples drawn from $P(\mathbb{X})$, up to the representational limits of the architecture. As a result, the matching loss naturally forms a pseudo-density or pseudo-indicator signal ("pseudo": it does not integrate to 1): regions frequently sampled from $P$ yield low predictor–prior discrepancy, while regions encountered rarely or not at all yield higher discrepancy.

Optimizing $(\mathfrak{P})$ with $P_{\text{expert}}$ as $P(\mathbb{X})$ learns an expert detector. Wang et al. (2019) learns such a detector, in an offline manner, and formulates an RL reward from it. Albeit sound, it falls short of capturing the expert distribution in practice. We argue that what plagues the method is that: *(a)* the pseudo-density is learned entirely offline, and *(b)* it posits that we only have *positive* signal (guided by $P_{\text{expert}}$), closely resembling one-class classification or positive-unlabeled learning. In fact, for the overwhelming majority of the agent's learning lifespan, the agent has sub-optimal behavior. Its behavior (informed by $P_{\text{agent}}$—whether on-policy or off-policy[1]) could therefore be treated as *negative* signal. This design position, in the classification analogy, turns one-class into binary classification. This is a view adopted by adversarial methods that train a reward model as the discriminator of a GAN, which is a binary classifier. IL methods based on GANs optimize a JS-divergence between distributions $P_{\text{expert}}$ and $P_{\text{agent}}$, which suffers from mode collapse and vanishing gradients when the supports do not overlap. This leads to instability or failure to train effectively. As such, they require careful regularization, especially in off-policy learning. In this

---

[1]Under the off-policy regime, the $P_{\text{agent}}$ shorthand designates following the off-policy distribution $\beta$ resulting from sampling experiences uniformly, without loss of generality, from the replay buffer. In effect, $\beta$ is a mixture of past $\pi_\theta$ updates. The bigger the buffer capacity, the older the oldest policy in the mixture. In the on-policy setting, following $P_{\text{agent}}$ would simply mean following the policy $\pi_\theta$.

work, we instead build an adversarial training scheme from the problem of predicting random priors in $\mathbb{R}^m$. Not only do we descend the gradients of $\ell\big(f_\xi(x), f_\xi^\dagger(x)\big)$ on expert data, we also ascend its gradients on agent-generated data. By using $h_\xi$ as a shorthand for $x \mapsto \ell\big(f_\xi(x), f_\xi^\dagger(x)\big)$, we define the loss $L(\xi)$ that will be the foundation of our adversarial training procedure:

$$L(\xi) := \mathbb{E}_{x \sim P_{\text{expert}}}\big[h_\xi(x)\big] - \mathbb{E}_{x \sim P_{\text{agent}}}\big[h_\xi(x)\big] \quad (1)$$

We refer to $h_\xi$ as the potential function, or simply the potential. To sum up, minimizing the loss $L(\xi)$ above trains the potential function $h_\xi$ to assign low values to expert data and high values to agent data. Therefore, composing $h_\xi$ with a monotonically decreasing transform naturally inverts this trend—assigning high values to expert states and low values to agent states. This is exactly the behavior we seek for reward assignment. Accordingly, we define the reward directly from the learned potential $h_\xi$ as: $r_\xi(x) := \exp\big(-h_\xi(x)\big)$. This choice ensures positivity, bounds the reward between 0 and 1 (the pairing loss $\ell$ is non-negative), and sharpens the contrast between expert and agent behavior. We discuss the reward numerics in APPENDIX J.

Next, we show that $L(\xi)$ enjoys empirical concentration guarantees, thereby enabling the agent to provably close the gap with the expert. In addition, now that all the learning components have been introduced, we point the reader to ALGORITHM 1 for the complete algorithmic outline of NGT.

## 4.2. Reward Learning: Theoretical Grounding

In this section, (i) we show that the adversarial objective $L(\xi)$ (EQ 1), derived in SECTION 4.1, is equivalent to an objective grounded in OT theory. (ii) We also characterize how tightly its empirical estimate concentrates—in other words, how closely it approximates the *true* objective $L(\xi)$.

We define $H_\xi^\Lambda$ as the set of potential functions $h_\xi : x \mapsto \ell\big(f_\xi(x), f_\xi^\dagger(x)\big)$ that are $\Lambda$-Lipschitz. Formally, $H_\xi^\Lambda := \big\{h_\xi : \mathbb{X} \to \mathbb{R}_+; x \mapsto \ell\big(f_\xi(x), f_\xi^\dagger(x)\big) \mid |h_\xi(x) - h_\xi(x')| \le \Lambda\, d(x, x'), \forall x, x' \in \mathbb{X}\big\}$ where $d$ is a ground metric over the input space $\mathbb{X}$, and $\Lambda < +\infty$. In particular, we make the following design choice: to train our reward model, we restrict the search for a potential function $h_\xi$ that minimizes $L(\xi)$ to the case where $\Lambda = 1$. That is, we optimize EQ 1 over the space of 1-Lipschitz potentials. Using the formalism above: we look for a function $h_\xi \in H_\xi^1$ that is the infimum of $L(\xi)$.

We observe that:

$$\inf_{h_\xi \in H_\xi^1} L(\xi) = -\sup_{h_\xi \in H_\xi^1}\Big(\mathbb{E}_{x \sim P_{\text{agent}}}\big[h_\xi(x)\big]$$
$$- \mathbb{E}_{x \sim P_{\text{expert}}}\big[h_\xi(x)\big]\Big)$$
$$= -\text{EMD}(P_{\text{agent}}, P_{\text{expert}}) \quad (2)$$

where EMD is the earth mover's distance between the two distributions. It quantifies the dissimilarity between two distributions, calculating the total effort required to transform (or transport) one into the other ($P_{\text{agent}} \to P_{\text{expert}}$). EQ 2 shows that when we update $\xi$ to minimize $L(\xi)$, we update $\xi$ to *maximize* the EMD between the distributions. As a result, descending along the gradients of $L(\xi)$—while ensuring that $h_\xi \in H_\xi^1$—maximizes the discrepancies between $P_{\text{agent}}$ and $P_{\text{expert}}$. In the context of OT, the learned potential function ($h_\xi$) embodies the Kantorovich-Rubinstein duality by iteratively approximating the optimal *dual* potentials that define the EMD. This approach aligns with the essence of the dual formulation: encoding the *cost landscape* of the transport problem. While the dual form has two potentials (one per $\mathbb{E}[\cdot]$), we learn only one ($h_\xi$). Specifically, we use $h_\xi$ for the first expectation, and $-h_\xi$ for the second. This design choice[2] reduces the dual constraint (from the dual formulation of the EMD) into a 1-Lipschitz continuity constraint on the single potential—hence our choice to look for $h_\xi \in H_\xi^1$—thereby ensuring consistency with the primal transport problem.

**The method:** We name our method Noise-Guided Transport (NGT) because: (i) Minimizing $L(\xi)$ guides $f_\xi(x)$ toward the noise returned by the prior network $f_\xi^\dagger(x)$ on expert data while pushing it away on agent data; (ii) The resulting $h_\xi$ yields an OT cost landscape. **In summary:** NGT derives its reward from an OT cost landscape $h_\xi$ designed to accentuate the gaps between the expert and agent distributions $P_{\text{expert}}$ and $P_{\text{agent}}$. This landscape sharpens the contrast between their occupancies, making the expert signal easier to discern. The agent ($\pi_\theta$) optimizes its actions on this landscape, which steers its behavior toward the expert and in effect reduces the discrepancies that $h_\xi$ exploits.

Finally, we derive a **concentration bound** for its empirical estimate $\hat{L}(\xi)$, computed from finite samples drawn from $P_{\text{expert}}$ and $P_{\text{agent}}$. We present the full analysis and proof in APPENDIX F. The result shows that $\hat{L}(\xi)$ converges to its expected value $L(\xi)$ at an exponential rate, with deviation controlled by the Lipschitz constant of the potential and the diameter of the input space. This bound quantifies the sample efficiency of our method and ensures that the empirical loss provides a reliable approximation of the true objective, giving theoretical control over generalization from finite samples.

## 4.3. Reward Learning: Practical Execution

In order for the potential $h_\xi : x \mapsto \ell\big(f_\xi(x), f_\xi^\dagger(x)\big)$—central to our reward design (SECTION 4.1)—to satisfy

---

[2]A similar design choice was made by (Arjovsky et al., 2017) (the EMD is the Wasserstein-1 distance). In APPENDIX E, we discuss how NGT relates to the problem formulated by (Arjovsky et al., 2017).

the equivalence laid out in EQ 2 (SECTION 4.2), it must be 1-Lipschitz: $h_\xi \in H_\xi^1$. Thus, we now examine how the Lipschitz constant of $h_\xi$ is governed by the Lipschitz properties of its constituent functions; namely, the predictor and prior networks $f_\xi$ and $f_\xi^\dagger$, and the pairing function $\ell$.

**Theorem 4.1** (Lipschitz constant of $h_\xi$). *Let $\Lambda(\cdot)$ denote the Lipschitz constant of a given function. By construction, $h_\xi$ is $\Lambda(h_\xi)$-Lipschitz continuous w.r.t. a ground metric $d$ over $\mathbb{X}$ with, $\forall x_1, x_2 \in \mathbb{X}$:*

$$\Lambda(h_\xi) = \Lambda(\ell)\big(\Lambda(f_\xi) + \Lambda(f_\xi^\dagger)\big) \tag{3}$$

*as Lipschitz constant. [Proof provided in* APPENDIX *G; results directly from function composition.]*

Having characterized how the Lipschitz constant of the potential $h_\xi$ depends on those of $f_\xi$, $f_\xi^\dagger$, and $\ell$, we now turn to the practical question of how to ensure that the condition $h_\xi \in H_\xi^1$ is satisfied; that is, how to ensure that $\Lambda(h_\xi) \leq 1$ holds in practice over $\mathbb{X}$. The following subsections describe the specific design choices we make (*e.g.*, architecture) to enforce this property in our method. We refer the reader to the extensive ablation studies presented in APPENDIX N that corroborate those choices.

### 4.3.1. CONTROLLING THE VALUES OF $\Lambda(f_\xi)$ AND $\Lambda(f_\xi^\dagger)$

EQ 3 shows that $\Lambda(f_\xi)$ and $\Lambda(f_\xi^\dagger)$ compound additively, and their sum compounds multiplicatively with $\Lambda(\ell)$. Therefore, $\ell$ can still act as a soft gate or low-pass filter that could prevent occasional spikes upstream and thereby cause destructive updates of the reward model. Importantly, $\Lambda(\ell)$ is fixed and does not change with $\xi$ updates —although it could be made to follow a schedule or heuristic. The same goes for $\Lambda(f_\xi^\dagger)$, since the prior network is never updated after initialization. As a result, $\Lambda(\ell)$ and $\Lambda(f_\xi^\dagger)$ depend only on design choices, while $\Lambda(f_\xi)$ can be altered during training. In practice, we do not aim for the "perfect-1" Lipschitz constant $\Lambda(h_\xi)$, i.e. $h_\xi \in H_\xi^1$; We have found that it is empirically enough to tame its value and keep it "close enough to 1" so as to avoid surges and spikes. We use spectral normalization (SN) (Miyato et al., 2018) on every linear layer of the predictor $f_\xi$ and prior $f_\xi^\dagger$, which constrains their Lipschitz constant by maintaining unit spectral norm. It is then the choice of non-linearity[3] in $f_\xi$ and $f_\xi^\dagger$ that dictates how their Lipschitz constants deviates from 1.

Given that the prior $f_\xi^\dagger$ is frozen after being initialized, *how* the networks are initialized is crucial. We use orthogonal initialization (OI) (Saxe et al., 2013; Hu et al., 2020) in every layer , which ensures that the weight matrix is a norm-preserving linear transformation. It ensures an even spread

in the feature space, avoiding redundancies (reducing correlations) in output space. Since the prior network is never updated and is initialized with OI, it has two appealing traits. (i) The random priors—output embedding of $f_\xi^\dagger$—are the result of a full-rank map, and therefore maximally utilize the available dimensions $m$ in the output embedding. (ii) The linear layers already have unit singular values (orthogonal matrix). The highest singular value is therefore already 1: applying SN has no effect, and would not be needed for the prior network. So, with OI and reasonably linear-like activations (e.g. ReLU, LeakyReLU), $\Lambda(f_\xi^\dagger)$ should be close to unit. Using SN is however required for the predictor. Otherwise, $\Lambda(f_\xi)$ can adopt an erratic behavior as $\xi$ gets updated. Importantly, we did not need to regularize the predictor with a gradient penalty (GP) (Gulrajani et al., 2017), while it is needed for every single off-policy adversarial IL SOTA baseline (see SECTION 5)[4]. This makes NGT cheaper and faster than its counterparts on that front. Indeed, GP is more computationally expensive than SN. GP effectively doubles the cost of the backward pass, while SN only adds minimal overhead.

### 4.3.2. CONTROLLING THE VALUE OF $\Lambda(\ell)$

What about $\Lambda(\ell)$? An obvious choice of loss with $\Lambda(\ell) \leq 1$ is the $L_1$ loss, or better yet, the Huber loss with parameter $\delta = 1$. In fact, most of the losses borrowed from robust regression are 1-Lipschitz and would satisfy our regularity desideratum. The Huber loss turned out to be an excellent option, and it acted as default in our experiments. Notably, NGT allows for the comparison of the embeddings $f_\xi^\dagger$ and $f_\xi$ directly, but they could also be pipped through another map or transform, e.g. learned feature map (see discussion in APPENDIX D), or a non-learned deterministic proxy function. We have tried the latter by wrapping both output embeddings with a softmax. In effect, comparing the softmaxes of embeddings evaluates the similarity between the distribution of output units[5]. We have found that this design choice gave comparable results to the Huber loss, and could give a slight edge in certain tasks.

These losses yielded excellent results, as reported in SECTION 5, with the exception of the Humanoid tasks, where none of the above losses produced satisfactory performance. We expanded the capabilities of NGT by enabling the use

---

[3] Anil et al. (2019) discusses how the choice of activation impacts whether the function approximator can actually be Lipschitz continuous when one encourages it to be via regularization.

[4] Gradient regularization was shown to be necessary in such methods (Blondé et al., 2020), and also more recently in a general generative modeling context (Huang et al., 2024).

[5] There is however loss of information since the match is only *relative*. This may be a boon however, and allow the agent to focus on the relative importance of features, rather than their absolute magnitudes. Since the random prior vector is a near-orthogonal map of the input vector, much of the input's original structure is preserved in it. If the agent benefits from relative matching in the input space (e.g. coordination or locomotion task), then it should benefit from relative matching in the random prior embedding.

of distributional losses—specifically, the histogram loss, "Gaussian type"—which were originally introduced for value learning in RL (Imani & White, 2018). We denote the loss with $\ell_{\mathrm{HLG}}$. It depends on four hyper-parameters, $(a, b, N, \sigma)$, where $[a, b]$ is the interval to partition into $N$ bins, and $\sigma$ dictates the spread of the Normal distribution involved in $\ell_{\mathrm{HLG}}$. We opted for $\ell_{\mathrm{HLG}}$ (regression $\to$ classification) because: *(i)* Spreading probability mass to neighboring locations reduces overfitting (label smoothing, (Szegedy et al., 2016)); *(ii)* Exploiting the ordinal structure of the regression enhances generalization across a range of target values; *(ii)* Classification losses have proved to produce better representations and have demonstrated greater robustness to non-stationarity. Turning regression into classification has enabled deep RL to finally reap benefits from scale (Hafner et al., 2023; Hansen et al., 2024; Farebrother et al., 2024). This aligns with the "scaling law" paper of Kaplan et al. (2020), which demonstrated how well cross-entropy scales. As we show in SECTION 5, $\ell_{\mathrm{HLG}}$ allowed us to successfully scale NGT to the Humanoid tasks.

Distributional losses such as $\ell_{\mathrm{HLG}}$ were originally introduced in the context of value learning, where the targets are scalar values and the model predicts a probability distribution over $N$ discrete bins. In our case, we repurpose this loss for reward learning, where the learning signal comes from predicting a vector of $m$-dimensional random priors. As a result, the prediction is effectively over $N \times m$ bins, rather than $N$ bins. This induces an architectural asymmetry between $f_\xi$ and $f_\xi^\dagger$: while the prior network $f_\xi^\dagger$ returns $m$ scalar targets, the predictor $f_\xi$ must now produce an output of size $N \times m$, representing a distribution over bins for each prior dimension. No other loss considered in this work introduces such an asymmetry. We describe the mechanism of $\ell_{\mathrm{HLG}}$ in detail in APPENDIX B, including how we extend the original loss to handle the extra dimension $m$ introduced by our method.

### 4.3.3. FROM CODE TO BOUND: DERIVING $\Lambda(\ell_{\mathrm{HLG}})$ VIA IMPLEMENTATION ANALYSIS

We want to determine $\Lambda(\ell_{\mathrm{HLG}})$, the Lipschitz constant of the $\ell_{\mathrm{HLG}}$ pairing loss. In particular, an insightful bound would reveal how the Lipschitz constant of $\ell_{\mathrm{HLG}}$ depends on its hyper-parameters, especially $\sigma$ and $N$, which together control the label smoothing capabilities of the loss. While Imani & White (2018), who introduced this loss, derived a local Lipschitz continuity bound with respect to the *parameters*, we derive ours with respect to the *inputs*. Both their findings and ours point to the same insight: the gradients are well-behaved. The part of their bound that concerns the Lipschitz continuity with respect to the inputs is simply the absolute bin-wise difference between predicted probability and transformed target bin value. We go further than that stage, considering the worst-case scenario, and

proceeding until the bound can only be expressed with the hyper-parameters of $\ell_{\mathrm{HLG}}$.

Based on our implementation of the $\ell_{\mathrm{HLG}}$ loss—see SNIPPET 1 in APPENDIX C, we want to upper bound the Lipschitz constant of the loss (w.r.t. its inputs). In the interest of space, we provide the intermediary results and their derivations in APPENDIX H. We first establish the theoretical framework by formalizing the elements involved in the snippet of the loss. We carry this out in DEF H.1. After establishing the groundwork, we present our core theoretical result, TH H.2, which expresses the Lipschitz constant of $\ell_{\mathrm{HLG}}$ in terms of the maximal reachable probability mass $p_{\max}$. We then develop a lemma LEM H.3, in which we derive an estimate of $p_{\max}$, which ultimately enables us to express $\Lambda(\ell_{\mathrm{HLG}})$ with respect to the hyper-parameters of the loss only, in TH 4.2.

**Theorem 4.2** (Lipschitz continuity of $\ell_{\mathrm{HLG}}$). *The histogram loss "Gaussian type" $\ell_{\mathrm{HLG}}$ is $\Lambda$-Lipschitz continuous with respect to the logits, with a Lipschitz constant that verifies the inequality:*

$$\Lambda \le \sqrt{1 + \left(\frac{C}{\sigma}\right)^2} \tag{4}$$

*where $C \coloneqq \Delta s \sqrt{(N-1)/(2\pi)}$. $\Delta s$ is the bin width (introduced in DEF H.1). Note, this result is subject to the approximation considerations from LEM H.3. Refer to APPENDIX H for greater details.*

**Discussion of guarantees and implications.** Theorem TH 4.2 shows how the Lipschitz constant of the function $x \mapsto \ell_{\mathrm{HLG}}(x, t)$ depends on $\sigma$. As $\sigma \to +\infty$, $C/\sigma \to 0$, so the upper bound on $\Lambda$ approaches 1. As $\sigma \to 0$ however, $C/\sigma \to +\infty$. This tells us that, when the Normal distribution is extremely narrow, the Lipschitz constant of $\ell_{\mathrm{HLG}}$ can grow unbounded. This translates to high sensitivity and therefore poor stability. As such, TH 4.2 advises for the use of a $\sigma$ value that is *high enough* with respect to the number of bins and the interval bounds $a$ and $b$ to prevent unsteadiness.

## 5. Experiments

We now evaluate the sample-efficiency and stability of NGT against a set of baseline methods across a suite of continuous control environments. We begin by describing the experimental setup, then present the baselines used for comparison. Finally, we report the results and discuss their implications.

All methods were re-implemented from a shared SAC-based actor-critic backbone, differing only in how the reward is computed or learned. APPENDIX L details the modifications made to DiffAIL (Wang et al., 2023), where we adapt the original reward implementation to ensure numerical stability. The general network architecture used across methods

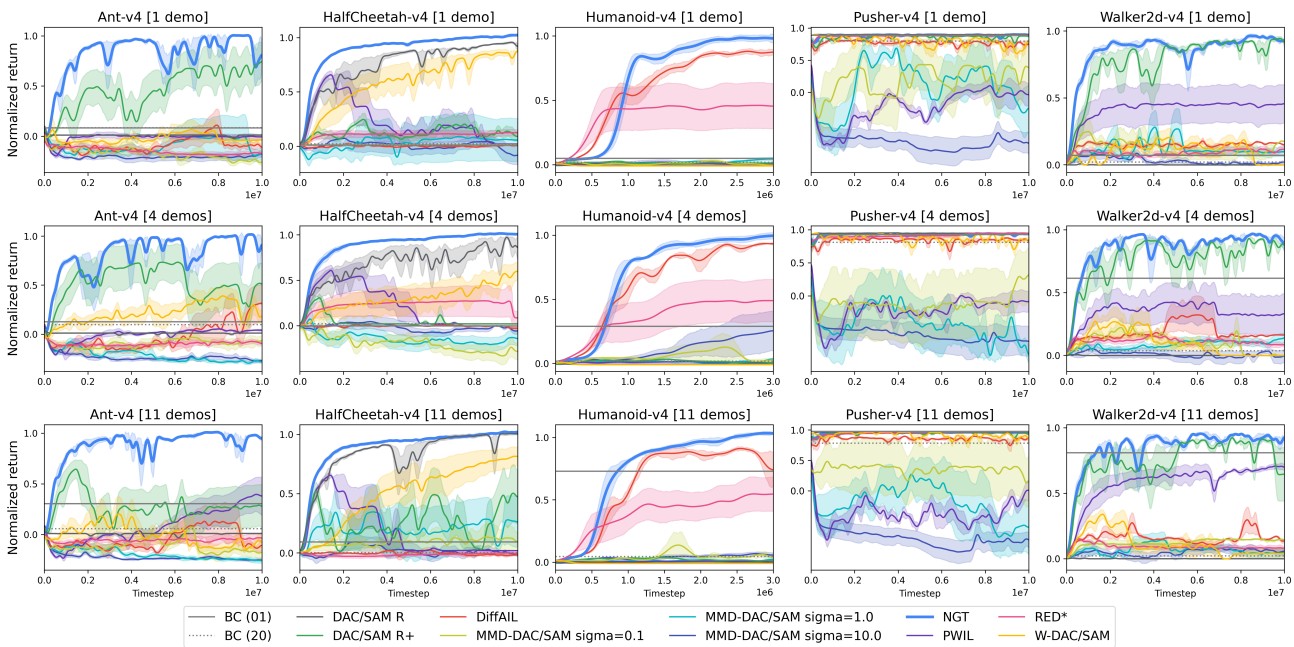

*Figure 2.* Performance comparison over various environments and numbers of demonstrations.

is shown in APPENDIX I. We share the hyperparameters in APPENDIX L. We compare NGT to baselines across environments with varying numbers of expert demonstrations (1, 4, and 11), subsampled at a rate of 20 with varying starting points $\in [0..19]$, as in (Ho & Ermon, 2016; Kostrikov et al., 2019; Blondé & Kalousis, 2019; Dadashi et al., 2021). Under that setting, 1 demonstration comprises 50 transitions (since they originally are composed of 1000 each). We use vectorized environments, with 4 parallel executors. When one step is carried out, we increment the counter by 4, reflecting the *true* number of interactions with *an* environment. Our experts are policies trained with SAC in the same environment as the agent, but using a different random seed. Each experiment is run with 4 random seeds. Importantly, during evaluation, a new seed is sampled from the initial one given to the agent at each episode reset, ensuring that each evaluation episode uses a different environment instance. This setup prevents the agent from memorizing trajectories and encourages genuine generalization toward expert behavior. We tackle the Gymnasium continuous control suite (Towers et al., 2024), whose complexity culminates with `Humanoid-v4`. We report the dimensions of the state and action spaces in APPENDIX M.

As tensor software, we use PyTorch (Paszke et al., 2019) and CUDA Graphs (APPENDIX K). CUDA Graphs enabled up to a 3x speedup for all algorithms; $\approx 5$ hours of training time for a humanoid on GPU. Each method tested fits within the memory of an NVIDIA RTX 4090 card, except DiffAIL for which we suggest reducing the replay buffer capacity by 1M to fit (required for `Humanoid-v4`).

We consider baselines that are either offline, or online off-policy. BC operates pure supervised learning on expert pairs (offline). We show two variants: BC with with the same subsampling as the other methods, and BC *without* any—we simply call the latter `BC(1)`. We find this adds perspective on *how data sparsity and scarcity impact the performance*. For example, `BC(1)` performs very well on `Humanoid-v4` with 11 demonstrations (11K pairs to train on). So, when expert data is abundant, BC is indeed a good option. Another method is PWIL (Dadashi et al., 2021) which iteratively solve a procedure aimed at minimizing the EMD in its *primal* form. As such, it *computes* a reward; it does not *learn* one. Next, we group DAC (Kostrikov et al., 2019) and SAM (Blondé & Kalousis, 2019) under a common framework. Both methods train their reward function using a JS-GAN discriminator. We refer to this combined baseline as `DAC/SAM` in the plots. We show two variants: `R+` uses $-log(1 - D)$ as reward ($\in \mathbb{R}_+$), and `R` uses $-log(1 - D) + log(D)$ ($\in \mathbb{R}$). `W-DAC/SAM` turns the JS-GAN discriminator into a WGAN critic (Arjovsky et al., 2017), and `MMD-DAC/SAM` replaces the EMD-maximizing objective of the WGAN critic with an MMD divergence (RBF, $\sigma \in \{0.1, 1, 10\}$) (Li et al., 2017; Xiao et al., 2019). We replicate RED (Wang et al., 2019) by only using the left term of $L(\xi)$ in NGT. The method is denoted `RED⋆`, where the $\star$ signifies that we apply an adaptive reward numerics scheme (APPENDIX J) instead of tuning a temperature parameter per environment like the original RED does. Finally, DiffAIL relies on a diffusion/de-noising task and uses the diffusion error in place of the discriminator output in

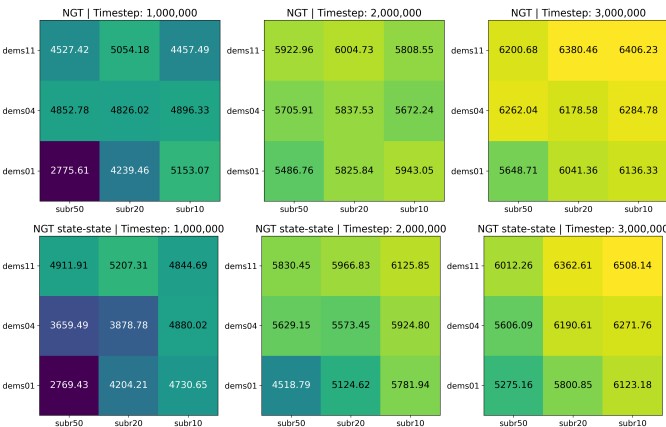

*Figure 3.* NGT's unnormalized performance across varying numbers of demonstrations and subsampling rates, in the *state-action* (first row) and *state-state* setting (second row), in `Humanoid-v4`.

a JS-GAN objective. Crucially, *only the reward model is diffusion-based*. Since the authors do not treat a `Humanoid` in the environment-specific configuration files of their codebase, we started our search from the hyper-parameters they used for `Ant`. Specifically, we use a gradient penalty coefficient of $0.1$ for DiffAIL, and the recommended $10$ for the other dual adversarial methods. Note, NGT did not require a gradient penalty regularizer outside spectral normalization, which is something DAC/SAM could not get away with, as show in (Blondé et al., 2020). We hypothesize that this effect stems from the greater numerical stability and smoother learning dynamics enabled by the potential function used in our reward learning design. Unless stated otherwise, all reported *returns are normalized per task* such that a score of 0 corresponds to the performance of a random agent, and 1 to that of the expert. Returns below the random baseline then yield negative scores. **Figures:** We present the main results in FIGURE 2, where we compare NGT to the baselines. In FIGURE 3, we further examine how NGT performs across finer-grained settings, varying both the number of demonstrations and the subsampling rate, in both the state-action and state-state scenarios. FIGURES 2 and 3 report results from 720 and 72 experiments, respectively.

**Interpretation of the observed results:** Overall, FIGURE 2 shows that NGT achieves expert performance across the board and outperforms the baselines. DiffAIL shows strong performance on `Humanoid-v4`, seemingly leveraging the high representational power of its diffusion model. Yet, it struggles in most of the others. Deeper per-environment tuning might make DiffAIL perform better, but we did not carry out per-task tuning for any method. FIGURE 3 shows that NGT exhibits consistent and stable scaling—even in the state-state setting. The figures also show that optimizing an EMD is not the whole story, or at least that it is not easy to estimate an EMD properly, judging by the difference in results between NGT and WGAN. Unlike binary classi-

fication—which may become trivial early in training and require strong regularization like gradient penalization—the $m$-dimensional prior prediction task scales more gracefully with model capacity. The histogram loss $\ell_{\text{HLG}}$ has the ability to apply $\sigma$-controlled label smoothing. In particular, since the pairing loss appears *twice* in the reward loss, we have the option to use different $\sigma$'s on each side. To emulate the typical JS-GAN trick of smoothing only the expert-side labels, we can use a higher $\sigma$ value in the expert-side expectation of $L(\xi)$. Furthermore, we support the design choices made in NGT through a series of **ablation studies** presented in APPENDIX N, and extend our analysis to **extra environments** in APPENDIX O. FIGURE 4 illustrates NGT's performance over 10 seeds, highlighting a tight variance across runs. Finally, we report a comparison of baseline **speeds** in APPENDIX P.

## 6. Conclusion

In this work, we develop Noise-Guided Transport (NGT), a sample-efficient imitation learning method in the low-data regime, where only a handful of demonstrations are available. The reward learning objective of NGT builds on prediction from random priors, and optimizes a distance rooted in optimal transport theory. Moreover, by leveraging distributional losses, NGT succeeds in learning to reproduce humanoid gaits, with as few as 20 transitions, even when actions are unavailable. It outperforms all baselines, and does not require gradient penalization. An intriguing direction for future work is to investigate the applicability of this objective to general generative modeling tasks. Beyond these technical contributions, we hope this work helps to re-spark interest in imitation learning under data-limited settings, enabling progress in applied domains such as biorobotics and healthcare, where demonstrations are extremely scarce and demand special care and considerations.

## Impact Statement

This paper introduces a data-efficient imitation learning method for continuous control, with the aim of improving performance in low-data settings. Potential positive impacts include more efficient learning for robotics and other control systems; potential risks are those typical of improved autonomy and should be managed by downstream users. We do not use human data and do not foresee direct negative societal impacts beyond these general considerations.

## Acknowledgements

This work was partially supported by the Swiss National Science Foundation through the METATHESIS project: "Modelling pathological gait with machine learning for treatment selection support" (grant no. 220435). Most of the experiments were run on the Baobab high-performance computing cluster at the University of Geneva.

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

# A. Algorithm

In ALGORITHM 1, the learned reward takes the triplet $(s_t, a_t, s_{t+1})$ as input to signify that, depending on the setting, the reward could be trained to use any of the following input combinations: $(s_t, a_t)$, $(s_t, s_{t+1})$, or $s_t$.

---

**Algorithm 1** Noise-Guided Transport (NGT) Algorithm (as presented in this work)

---

1: Initialize parameters of policy network $\pi_\theta$, Q-networks $Q_{\omega_1}$, $Q_{\omega_2}$, target Q-network parameters $\bar{\omega}_1 \leftarrow \omega_1$, $\bar{\omega}_2 \leftarrow \omega_2$, and reward model $r_\xi$
2: Initialize temperature parameter $\alpha$ and target entropy $\mathcal{H}$
3: Initialize replay buffer $\mathcal{D}$ and expert demonstration dataset $\mathcal{E}$
4: **for** each iteration **do**
5:     **for** each environment step **do**
6:         Sample action $a_t \sim \pi_\theta(a_t|s_t)$
7:         Execute $a_t$ in environment, observe $s_{t+1}$
8:         Store $(s_t, a_t, s_{t+1})$ in $\mathcal{D}$
9:         `// not storing rewards`
10:     **end for**
11:     **for** each gradient step **do**
12:         `// Update reward model`
13:         Sample a minibatch of transitions $(s_t, a_t, s_{t+1})$ $\mathcal{B}_{\mathcal{E}}$ from expert dataset $\mathcal{E}$
14:         Sample a minibatch of transitions $(s_t, a_t, s_{t+1})$ $\mathcal{B}_{\mathcal{D}}$ from replay buffer $\mathcal{D}$
15:         Update reward model parameters $\xi$ by minimizing:

$$L(\xi) = \frac{1}{|\mathcal{B}_{\mathcal{E}}|} \sum_{t \in \mathcal{B}_{\mathcal{E}}} h_\xi(s_t, a_t, s_{t+1}) - \frac{1}{|\mathcal{B}_{\mathcal{D}}|} \sum_{t \in \mathcal{B}_{\mathcal{D}}} h_\xi(s_t, a_t, s_{t+1})$$

16:         `// Update actor-critic`
17:         Sample a minibatch of transitions $(s_t, a_t, s_{t+1})$ from $\mathcal{D}$
18:         Compute target Q-value using reward model $r_\xi$:

$$y_t = r_\xi(s_t, a_t, s_{t+1}) + \gamma \mathbb{E}_{a_{t+1} \sim \pi_\theta} \left[ \min_{i=1,2} Q_{\bar{\omega}_i}(s_{t+1}, a_{t+1}) - \alpha \log \pi_\theta(a_{t+1}|s_{t+1}) \right]$$

19:         Update Q-function parameters $\omega_i$ by minimizing:

$$L(\omega_i) = \frac{1}{|\mathcal{B}_{\mathcal{D}}|} \sum_{t \in \mathcal{B}_{\mathcal{D}}} (Q_{\omega_i}(s_t, a_t) - y_t)^2 \quad \text{for } i = 1, 2$$

20:         Update policy parameters $\theta$ by minimizing:

$$L(\theta) = \frac{1}{|\mathcal{B}_{\mathcal{D}}|} \sum_{t \in \mathcal{B}_{\mathcal{D}}} \mathbb{E}_{a_t \sim \pi_\theta} \left[ \alpha \log \pi_\theta(a_t|s_t) - \min_{i=1,2} Q_{\omega_i}(s_t, a_t) \right]$$

21:         Adjust temperature $\alpha$ (optional) by minimizing:

$$L(\alpha) = -\frac{1}{|\mathcal{B}_{\mathcal{D}}|} \sum_{t \in \mathcal{B}_{\mathcal{D}}} \alpha \left( \log \pi_\theta(a_t|s_t) + \mathcal{H} \right)$$

22:         Update target Q-network parameters:

$$\bar{\omega}_i \leftarrow \tau \omega_i + (1 - \tau) \bar{\omega}_i \quad \text{for } i = 1, 2$$

23:     **end for**
24: **end for**

---

# B. HL-Gaussian Loss: Mechanism and How We Extend It

In essence, $\ell_{\text{HLG}}$ maps the scalar target onto $N$ bins (spread evenly across $[a, b]$) using a transformation that assigns the highest probability mass to the bin containing the scalar, while distributing the remaining mass to neighboring bins in the shape of a bell curve. This redistribution of mass to adjacent locations is akin to target smoothing. The model being trained with this loss predicts one value per bin, i.e. a $N$-dimensional output. Those logits are transformed into probabilities with a softmax layer. The two vectors of size $N$ are then compared using cross-entropy. To compare $f_\xi$ to $f_\xi^\dagger$, we therefore introduce *asymmetry* in their architecture. The random priors are still in $\mathbb{R}^m$, but the predictor returns an embedding in $\mathbb{R}^{m \times N}$ (column vector). It is then rearranged into a $N \times m$ matrix ($\in \mathcal{M}_{m,N}(\mathbb{R})$). After softmax-ing row-wise, each now of this matrix is a vector whose $N$ elements are interpreted as predicted probabilities over bins. On the target side, the $m$ scalar random priors returned by $f_\xi^\dagger$ are each transformed into a probability vector spanning $N$ bins. In effect, we now have a $\mathcal{M}_{m,N}(\mathbb{R})$ matrix from $f_\xi^\dagger$ and from $f_\xi^\dagger$. Finally, the matrices are compared row-wise using $N$-bin cross-entropy. Our implementation of the $\ell_{\text{HLG}}$ loss is given in SNIPPET 1, in APPENDIX C,

# C. HL-Gaussian Loss: Code Snippet

In this section, we provide a PyTorch (Paszke et al., 2019) code snippet of our augmented implementation of the HL-Gaussian loss (Imani & White, 2018). Our code builds on the reference snippet shared in the appendix of Farebrother et al. (2024), with several modifications. One change that significantly improved numerical stability was the addition of a small constant $\epsilon = 10^{-6}$ to the denominator in the `transform_to_probs` function, as shown in SNIPPET 1. Matrix reshaping into $\mathcal{M}_{m,N}(\mathbb{R})$ is carried out using the `einops` library.

*Listing 1.* HL-Gaussian loss

```python
from einops import rearrange
import torch
from torch import nn
from torch.nn import functional as ff

class HLGaussLoss(nn.Module):

    def __init__(self,
                 *,
                 min_value: float,
                 max_value: float,
                 num_bins: int,
                 sigma: float,
                 device: torch.device,
                 reduction: str = "none"):
        super().__init__()
        self.min_value = min_value
        self.max_value = max_value
        self.num_bins = num_bins
        self.sigma = sigma
        self.device = device
        self.reduction = reduction
        self.support = torch.linspace(
            min_value, max_value, num_bins + 1, dtype=torch.float, device=self.device)
        self.sqrt_of_two = torch.sqrt(torch.tensor(2.0, device=self.device))

    def forward(self, logits: torch.Tensor, target: torch.Tensor) -> torch.Tensor:
        logits = rearrange(logits, "b (c d) -> b c d", c=self.num_bins)
        target_probs = self.transform_to_probs(target)
        target_probs = rearrange(target_probs, "b d c -> b c d", c=self.num_bins)
        return ff.cross_entropy(logits, target_probs, reduction=self.reduction)

    def transform_to_probs(self, target: torch.Tensor) -> torch.Tensor:
        operand1 = self.support - target.unsqueeze(-1)
        operand2 = self.sqrt_of_two * self.sigma
        operand = operand1 / operand2
        cdf_evals = torch.special.erf(operand)
        z = cdf_evals[..., -1] - cdf_evals[..., 0]
        bin_probs = cdf_evals[..., 1:] - cdf_evals[..., :-1]
        return bin_probs / (z + 1e-6).unsqueeze(-1)

    def transform_from_probs(self, probs: torch.Tensor) -> torch.Tensor:
        centers = (self.support[:-1] + self.support[1:]) / 2
        return torch.sum(probs * centers, dim=-1)
```

## D. Related Works (Expansion)

In this section, we expand upon the related work discussed in SECTION 3.

The structure of the loss we train our reward with, $\ell(f(x), f^\dagger(x))$, echoes the loss used for contrastive learning in self-supervised learning, $\ell(f(x), f(y))$ (*e.g.*, in SimCLR (Chen et al., 2020)). However, unlike contrastive learning, where $y$ represents an augmented or semantically similar version of $x$, our formulation leverages $f^\dagger(x)$ as a prior/reference signal.

Concentration bounds in optimal transport (OT) provide guarantees on the convergence rate of empirical measures in Wasserstein distance (also called earth-move distance, EMD). (Fournier & Guillin, 2015) establishes explicit bounds on the Wasserstein distance, with rates that depend on the dimensionality of the space. Their results are particularly sharp in the one-dimensional case, such as traditional EMD. (Villani, 2009) serves as a comprehensive reference for OT theory, including Wasserstein concentration bounds. While Villani does not focus specifically on empirical concentration inequalities, key theoretical tools such as the Lipschitz constant and McDiarmid's inequality appear in derivations, like in ours. More recently, (Weed & Bach, 2019) provides a detailed analysis of finite-sample convergence rates, refining previous results and offering insights into high-dimensional settings. These add perspective to the concentration guarantees we derived in this work.

The influence of the Lipschitz constant on the reward function has been investigated in depth in (Blondé et al., 2020). The authors argue, using DAC (Kostrikov et al., 2019) and SAM (Blondé & Kalousis, 2019) as baselines, that gradient penalization (Gulrajani et al., 2017) is necessary for learning in the off-policy setting, as spectral normalization (Miyato et al., 2018) alone proves insufficient. In contrast, NGT (this work) succeeds with *only* spectral normalization. How to enforce Lipschitz-continuity in neural networks and how it impact their performance has also been tackled in (Khromov & Singh, 2024). Earlier in SECTION 4, we made a connection between RED (Wang et al., 2019) and one-class classification (positive-unlabeled, PU learning) as each posits only positive signal is available. PU learning was used in adversarial IL in (Hou et al., 2018) but resulted in subpar performance (Blondé et al., 2020).

DreamerV3 (Hafner et al., 2023) and TD-MPC2 (Hansen et al., 2024) are other works that incorporate classification losses for value learning in RL, though not as their primary focus. In contrast, Farebrother et al. (2024) explicitly centers on this idea, arguing that using classification losses for value learning enables RL to benefit from scale. Our findings support and extend their observations, in that we show that classification losses can also play a crucial role in reward learning, in an imitation learning context.

Reinforcement learning from human feedback (RLHF, (Christiano et al., 2017)) has seen a monumental resurgence in recent years with the rise of conversational agents built from large language models (LLMs). RLHF is used in LLM post-training to align the agent with human incentives through a reward model. **Modern RLHF techniques align with our reward learning logic**: perform gradient descent on preferred behaviors and gradient ascent on undesired ones, **as in Direct Preference Optimization (DPO)** (Rafailov et al., 2023). At the intersection of reward Lipschitz continuity and LLMs, WARM (Ramé et al., 2024) highlights the critical role of strict reward regularity in ensuring the effectiveness of a reward model for LLM post-training. Their findings closely align with those of (Blondé et al., 2020), which we previously discussed. Finally, a recent trend in the field views RLHF through the lens of inverse reinforcement learning (IRL), framing the reward model as an implicit representation of human preferences (Wulfmeier et al., 2024; Sun, 2024). **"RLHF as *adversarial* IRL" may soon emerge as a key direction in the field.**

Outside the scope of RL, Lemos et al. (2023) created a technique based on random points to test the accuracy of posterior estimators. The technique is intended for evaluation rather than model training.

## E. WGAN

In the WGAN (Arjovsky et al., 2017) formulation, the potential function corresponds directly to the critic, and the generator is trained by gradient descent on the learned potential. By contrast, we train our actor-critic architecture via policy gradient, with a reward constructed from the learned potential $h_\xi$. Although both NGT and WGAN optimize an EMD, they rely on fundamentally different potential functions. Crucially, the behavior of these potentials can significantly affect learning dynamics and training stability, as demonstrated by our experimental results in SECTION 5, where the `W-DAC`/`SAM` baseline implements the WGAN potential. Notably, while the WGAN potential (i.e. the critic) takes value in $\mathbb{R}$, our potential $h_\xi$ returns values in $\mathbb{R}_+$. What's more, while the WGAN critic is unconstrained over $\mathbb{R}$, the values returned by our predictor network $f_\xi$ are **implicitly anchored** by those of the prior network $f_\xi^\dagger$, which prevents $h_\xi$ from attaining excessively large values in $L(\xi)$.

# F. Concentration of Empirical Objective: Theoretical Results and Proofs

The loss $L(\xi)$ derived in SECTION 4.1 to learn a robust reward signal is defined in EQ 1 as:

$$L(\xi) = \mathbb{E}_{x \sim P_{\text{expert}}}\big[h_\xi(x)\big] - \mathbb{E}_{x \sim P_{\text{agent}}}\big[h_\xi(x)\big]$$

where $h_\xi$ is 1-Lipschitz *w.r.t.* a ground metric over the input space $\mathbb{X}$: $d(x, x')$, $\forall x, x' \in \mathbb{X}$. To support the reliability of this objective, **we set out to derive a concentration bound for its empirical estimate** $\hat{L}(\xi)$, computed from finite samples drawn from $P_{\text{expert}}$ and $P_{\text{agent}}$. The diameter, for the input space $\mathbb{X}$ and ground metric $d$, is defined as $\mathrm{diam}(\mathbb{X}) := \sup_{x,x' \in \mathbb{X}} d(x, x')$. In what follows, We omit the "$\xi$" subscripts to lighten the notations ($h_\xi \to h$). Also, we consider the $h$ functions that are $\Lambda$-Lipschitz: $h \in H^\Lambda$ ($H_\xi^\Lambda \to H^\Lambda$). We treat the case $\Lambda = 1$ in a corollary. Finally: $L(\xi) \to L$.

**Assumption F.1.** $\Lambda > 0$ and $\mathrm{diam}(\mathbb{X}) < +\infty$.

**Theorem F.2** (Concentration bound for the reward loss)**.** *Let $X^e = \{x_1^e, \ldots, x_n^e\}$ and $X^a = \{x_1^a, \ldots, x_n^a\}$ be sets of $n$ independent samples drawn from $P_{\text{expert}}$ and $P_{\text{agent}}$. Let $h \in H_\Lambda$, and let the empirical loss $\hat{L}$ be defined as:*

$$\hat{L} := \frac{1}{n}\sum_{i=1}^{n} h(x_i^e) - \frac{1}{n}\sum_{j=1}^{n} h(x_j^a) \tag{5}$$

*Then, the **deviation** of the empirical loss $\hat{L}$ (EQ 5) from its expected value $L$ (EQ 1) verifies:*

$$\mathbb{P}\big(|\hat{L} - L| \geq \epsilon\big) \leq \exp\left(-\frac{\epsilon^2 n}{\Lambda^2 \, \mathrm{diam}(\mathbb{X})^2}\right) \tag{6}$$

The proof of TH F.2 relies on McDiarmid's method of bounded differences (McDiarmid, 1989).

It proceeds as follows.

*Proof.* First, we conduct a sensitivity analysis of $\hat{L}$ by evaluating the **maximum change** in $\hat{L}$ when a single sample is substituted. Starting with the first term of $\hat{L}$, we see that a replacement $x_i^e \in X^e \to x_i^{e'}$, without loss of generality causes the change:

$$\left|\frac{h(x_i^e)}{n} - \frac{h(x_i^{e'})}{n}\right| = \frac{1}{n}\left|h(x_i^e) - h(x_i^{e'})\right| \tag{7}$$

Since this applies for any replacement in the first term of $\hat{L}$, we can try to upper bound the term above with a bound that does not depend on the indices of the samples, and that entity would then *bound all the differences* in the term.

$$\frac{1}{n}\left|h(x_i^e) - h(x_i^{e'})\right| \leq \frac{\Lambda}{n}d(x_i^e, x_i^{e'}) \leq \frac{\Lambda}{n}\,\mathrm{diam}(\mathbb{X}) \tag{8}$$

The first transition is due to $h$ being $\Lambda$-Lipschitz continuous by assumption. The second applies the definition of diameter. By symmetry, the sensitivity is the same for every replacement in the second term of $\hat{L}$. Due to all the differences being bounded, we can use McDiarmid's inequality (McDiarmid, 1989). To compute the bound, we need to compute the sum of squares of the bounds of the individual changes. Since there are $2n$ replacements and that we upper-bounded every replacement by an index-independent value $(\Lambda/n)\,\mathrm{diam}(\mathbb{X})$, the total sensitivity to insert in McDiarmid's bound is:

$$S := 2n\left(\frac{\Lambda}{n}\,\mathrm{diam}(\mathbb{X})\right)^2 = 2\frac{\Lambda^2}{n}\,\mathrm{diam}(\mathbb{X})^2 \tag{9}$$

We conclude by using the inequality with the calculated $S$:

$$\mathbb{P}\big(|\hat{L} - L| \geq \epsilon\big) \leq \exp\left(-\frac{2\epsilon^2}{S}\right) \tag{10}$$

The reduction of the operand yields the result in TH F.2. $\qquad\square$

A PAC-style bound (probably approximately correct) can easily be derived from TH F.2 by equaling the bound to a $\delta$ and reducing. We can also derive a **corollary** for the case "$h \in H^1$" $\Lambda = 1$.

**Corollary F.3** (For $h \in H^1$).

$$\mathbb{P}\big(|\hat{L} - L| \geq \epsilon\big) \leq \exp\left(-\frac{\epsilon^2 n}{\text{diam}(\mathbb{X})^2}\right) \tag{11}$$

The proof is immediate from TH F.2 by setting $\Lambda = 1$.

## G. Potential Function Lipschitz Continuity: Proof

We here provide a proof for the theorem TH 4.1, presented in SECTION 4.3 without proof.

This theorem characterizes the Lipschitz constant of the potential function $h_\xi$ in terms of the Lipschitz constants of the individual functions that composes it: $f_\xi$, $f_\xi^\dagger$, and $\ell$.

*Proof.* Let $\Lambda(\cdot)$ denote the Lipschitz constant of a given function. We aim to bound the Lipschitz constant of the composite function $h_\xi$. To do so, we derive an upper bound on the deviation of $h_\xi$ in terms of its constituent functions. By the properties of Lipschitz continuity under composition:

$$
\begin{aligned}
\big|h_\xi(x_1) - h_\xi(x_2)\big| &= \big|\ell\big(f_\xi(x_1), f_\xi^\dagger(x_1)\big) - \ell\big(f_\xi(x_2), f_\xi^\dagger(x_2)\big)\big| \\
&\leq \Lambda(\ell)\left(\left\|\big(f_\xi(x_1), f_\xi^\dagger(x_1)\big) - \big(f_\xi(x_2), f_\xi^\dagger(x_2)\big)\right\|\right) \\
&\leq \Lambda(\ell)\left(\big\|f_\xi(x_1) - f_\xi(x_2)\big\| + \big\|f_\xi^\dagger(x_1) - f_\xi^\dagger(x_2)\big\|\right) \\
&\leq \Lambda(\ell)\left(\Lambda(f_\xi)\, d(x_1, x_2) + \Lambda(f_\xi^\dagger)\, d(x_1, x_2)\right) \\
&\leq \Lambda(\ell)\big(\Lambda(f_\xi) + \Lambda(f_\xi^\dagger)\big)\, d(x_1, x_2)
\end{aligned}
\tag{12}
$$

$\forall x_1, x_2 \in \mathbb{X}$. Therefore, $h_\xi$ is Lipschitz continuous with constant:

$$\Lambda(\ell)\big(\Lambda(f_\xi) + \Lambda(f_\xi^\dagger)\big) \tag{13}$$

with respect to the ground metric $d$ over $\mathbb{X}$ with, $\forall x_1, x_2 \in \mathbb{X}$. $\qquad \square$

## H. HL-Gaussian Loss Lipschitz Continuity: Theoretical Results and Proofs

In this appendix, we present theoretical results and corresponding proofs demonstrating and characterizing the Lipschitz continuity of the HL-Gaussian loss function (Imani & White, 2018), introduced in SECTION 4.3.2.

**Definition H.1** (Groundwork). Let $\{s_0, \ldots, s_N\}$ be a partition of the interval $[a, b]$ into $N$ bins. As such, $s_0 = a$, $s_N = b$, and each bin has width $\Delta s := s_{i+1} - s_i$, $\forall i \in [0, N-1] \cap \mathbb{N}$. The general definition of probability for bin $i$, under the Normal distribution, is, $\forall t \in [a, b]$:

$$p_i(t) := \frac{\Phi_0\big(\frac{s_{i+1}-t}{\sigma}\big) - \Phi_0\big(\frac{s_i-t}{\sigma}\big)}{\Phi_0\big(\frac{b-t}{\sigma}\big) - \Phi_0\big(\frac{a-t}{\sigma}\big)} \tag{14}$$

where $\Phi_0$ is the CDF of the standard normal distribution. The denominator ensures that if there were probability mass of the $t$-centered Gaussian to be put outside $[a, b]$ (then *truncated*), the bins would be rebalanced by being uniformly attributed the extra mass needed for the $p_i$'s to sum up to 1. $\Phi_0$ can be expressed with the *special function* erf. Hence:

$$p_i(t) = \frac{\text{erf}\big(\frac{s_{i+1}-t}{\sqrt{2}\sigma}\big) - \text{erf}\big(\frac{s_i-t}{\sqrt{2}\sigma}\big)}{\text{erf}\big(\frac{b-t}{\sqrt{2}\sigma}\big) - \text{erf}\big(\frac{a-t}{\sqrt{2}\sigma}\big)} \tag{15}$$

which is how the *target transformation* is operated for a scalar $t \in [a, b]$ in the code snippet CODE 1 in APPENDIX C.

Let $\tilde{x} := (x_0, \ldots, x_{N_1}) \in \mathbb{R}^N$ be a vector of *logits* (typically the output of neural net). Following, CODE 1 in APPENDIX C, we go from *predicted logits* to *predicted probabilities* with a **softmax** over the $\mathbb{R}^N$ vector (also summing up to one by construction). The predicted distribution is denoted by $q$. The per-bin probability mass is:

$$q_i(\tilde{x}) := \frac{\exp(x_i)}{\sum_{j=0}^{N-1} \exp(x_j)} \tag{16}$$

$\forall i \in [0, N-1] \cap \mathbb{N}$. Finally, we define $\ell_{\mathrm{HLG}}$ as the **cross-entropy** between the transformed target distribution $p(t)$ and the predicted distribution $q(\tilde{x})$. It is the bin-wise sum:

$$\ell_{\mathrm{HLG}}(\tilde{x}, t) := - \sum_{0}^{N-1} p_i(t) \log\left(q_i(\tilde{x})\right) \tag{17}$$

$\forall i \in [0, N-1] \cap \mathbb{N}$ and $\forall t \in [a, b]$.

**Theorem H.2** (Lipschitz constant of $\ell_{\mathrm{HLG}}$ ($p_{\max}$ version))**.** *For any vector of logits over bins $\tilde{x} \in \mathbb{R}^N$, and $\forall t \in [a, b]$, the loss $\ell_{\mathrm{HLG}}$ is $\Lambda$-Lipschitz continuous w.r.t. $\tilde{x}$, with:*

$$\Lambda \le \sqrt{1 + (N-1)p_{\max}^2} \tag{18}$$

*where $p_{\max}$ is the maximal probability mass reachable by $p(t)$ on a bin of its support $[a, b]$. It is achieved on the bin $k$ where the $t$ value falls in ($p_k(t) = p_{\max}$), and upper bounds the mass of any other bin: $\forall i \ne k, p_i(t) \le p_{\max}$.*

*Proof.* Starting from a known result about the gradient of the cross-entropy over discrete vectors with softmax, we can write the partial derivative of $\ell_{\mathrm{HLG}}$ w.r.t. $x_j$, which is:

$$\frac{\partial \ell_{\mathrm{HLG}}}{\partial x_j} = q_j(\tilde{x}) - p_j(t) \tag{19}$$

Hence:

$$\nabla_{\tilde{x}} \ell_{\mathrm{HLG}}(\tilde{x}, t) = \left(q_0(\tilde{x}) - p_0(t), \ldots, q_{N-1}(\tilde{x}) - p_{N-1}(t)\right) \tag{20}$$

The Lipschitz constant $\Lambda$ measures how large the gradient can get across all possible $\tilde{x} \in \mathbb{R}^N$ and $t \in [a, b]$, i.e. :

$$\Lambda = \sup_{\tilde{x}, t} \|\nabla_{\tilde{x}} \ell_{\mathrm{HLG}}(\tilde{x}, t)\|_2 = \sup_{\tilde{x}, t} \|q(\tilde{x}) - p(t)\|_2 \tag{21}$$

In order to find an upper bound on $\Lambda$, consider the *worst-case* scenario: when the distributions differ the most. The predicted distribution at $\tilde{x}$, $q(\tilde{x})$, is a one-hot vector, without loss of generality with $q_j = 1$ and $q_i = 0$ for any $i \ne j$. The transformed target distribution at $t$, $p(t)$, has maximum probability mass in a bin $k$, such that $p_k(t) = p_{\max}$. In the worst case, $j \ne k$.

We square the norm above, under this worst-case mismatch:

$$\|q(\tilde{x}) - p(t)\|_2^2 = \left(1 - p_j(t)\right)^2 + \sum_{i \ne j} \left(0 - p_i(t)\right)^2 \tag{22}$$

Since in this imagined scenario, $j \ne k$, $k$ is in the second term. We therefore upper bound each individual term of the sum by the term that has the highest value: the $k$-th one.

$$\sum_{i \ne j} p_i(t)^2 \le (N-1)p_k(t)^2 = (N-1)p_{\max}^2 \tag{23}$$

Because $j \ne k$, all we can do is upper bound the first term $(1 - p_j(t))^2$ by 1. This concludes with the final result. $\square$

**Lemma H.3** (Maximum probability mass $p_{\max}$)**.** *The maximum value $p(t)$ can take on a bin, $p_{\max}$, verifies:*

$$p_{\max} \approx \frac{\Delta s}{\sigma \sqrt{2\pi}} \tag{24}$$

*This approximation tends toward an equality as (1) the bin width $\Delta s$ gets smaller w.r.t. $\sigma$ ($\Delta s \ll \sigma$), and as (2) the interval $[a, b]$ in which $t$ is defined covers most of the Gaussian's probability mass (e.g. $[a, b] \supset [t - 3\sigma, t + 3\sigma]$).*

*Proof.* $p_{\max}$ denotes the maximum probability mass a bin takes. It is taken by $p(t)$ at the bin $t$ falls in. Say it is bin $k$, without loss of generality. The PDF of the $t$-centered Normal $\mathcal{N}(t, \sigma)$ takes value $1/\left(\sigma \sqrt{2\pi}\right)$ at $t$. If we assume that the

Gaussian is approximately uniform over the bin containing $t$—bin $k$—we can approximate the numerator of the target transformation $p_i(t)$ (as laid out in DEF H.1) with the *area of the rectangle*:

$$p_{\max} = p_k(t) \approx \frac{\Delta s / (\sigma \sqrt{2\pi})}{\Phi_0 \left( \frac{b-t}{\sigma} \right) - \Phi_0 \left( \frac{a-t}{\sigma} \right)} \tag{25}$$

The approximation of the **numerator** by the area of a rectangle becomes increasingly valid (closer to equality) as the bin width decreases *w.r.t.* the statistic $\sigma$. In addition, the normalization factor in the **denominator** approaches 1 as the interval $[a, b]$ covers the space where the Gaussian centered at $t$ would put probability mass. A good coverage would be ensured if $[t - 3\sigma, t + 3\sigma] \subset [a, b]$. $\qquad \square$

We now conclude the chain of theoretical results and proofs with TH 4.2, in the main text. The proof of TH 4.2 is straightforward, following directly from substituting the result of LEM H.3 into TH H.2.

## I. Network Architecture

The actor, critic, and reward networks all have two hidden layers of width 256 units. The actor and critic use ReLU activations, while the reward network *all* use LeakyReLU activations with leak 0.05. We did *not* use layer normalization (Ba et al., 2016) in *any* network, and used orthogonal initialization (Saxe et al., 2013; Hu et al., 2020) in every network. We used spectral normalization (Miyato et al., 2018) for every layer of the reward network.

We tried the asymmetric architecture proposed in (Nikulin et al., 2023), where predictor and prior networks have different architectures involving new ways of extracting features from state and action in a continuous control context—the context we consider in this work. Among the techniques in use are bilinear layers and FiLM (Perez et al., 2018). The authors seem the get benefits from the proposed changes in feature engineering for offline RL. We however have not seen any benefit from this architecture. Besides, it has a significant toll on computational complexity. **In this work, we have show that it is possible to use a signal learned from random priors to solve tasks in continuous control**, which is what their architecture was claimed to unlock.

## J. Reward Numerics

Since different losses $\ell$ lead to different scales, it can be hard to determine the effectiveness of a loss design simply because the scale and shift of the resulting rewards might disagree with the agent. In order to have a controlled environment in which $\ell$ choices can be compared with minimal confounding factors, we adopt a simple scaling and shifting scheme to the designed reward. It could rely on the mean and standard deviation of the reward over the mini-batch—with an inclusion of an exponential moving average (EMA) with configurable decay,but we went for the most robust option and used percentiles statistics instead. We were inspired by the *return* rescaling mechanism presented in DreamerV3 (Hafner et al., 2023). Percentiles are indeed more robust statistics (*e.g.*, against outliers) compared to mean and standard deviation. We operate as follows.

First we divide the batch of rewards $r$ by $\mathrm{Perc}_{0.95}(r) - \mathrm{Perc}_{0.05}(r)$, the gap between the 5th percentile of the batch and the 95th. Then, we shift the reward by $\mathrm{Perc}_{0.05}(r)$ to re-center. We do not use an EMA. Despite its potential stabilization benefits, it exposes the method to rely too much on older statistics, thereby slowing down the method relatively to the other approaches. Note, since the percentiles are computed on the batch, the batch size hyper-parameter can not be too small. Picking a batch size below 16 for example would be ill-advised and could lead to degenerate cases. On the flip side, such a scheme is well-adapted to vectorized environments, where data collection is parallelized, and the batch size can usually be scaled up. We do not use a temperature hyper-parameter in the exponential of the reward.

In RND (Burda et al., 2018), the authors divide the operand of the exponential (like in our case, used to turn the negative loss into a positive reward signal) by a running average of the standard deviation. This statistic plays of the role of temperature. We hypothesize that this technique works well for RND because the reward is used as a reward bonus, aiming at better RL exploration. The method does not care too much about which of two novel states is the most novel, as long as the agent does explore novel states. It is however of primary importance in our case, which could explain why this technique did not lead to good results in our early experiments.

In RED (Wang et al., 2019), the authors use a different hard-coded temperature for every environment. The temperature

ranges from $\tau = 250$ to $\tau = 250,000$. In SECTION 5, we show results for our implementation of RED, and we apply the adaptive reward treatment described above instead of an unfair tuning of temperature $\tau$ per environment.

## K. CUDA Graphs

We use CUDA Graphs[6] in all the algorithms ran in the context of this work. CUDA Graphs optimize GPU workloads by capturing a **static** sequence of operations (*e.g.*, computations and memory transfers) into a *graph* that can be *replayed* with minimal overhead. **We generally gained a *3x speedup* on every workflow**, with extra precaution taken to ensure our PyTorch computational graphs were *static*, *i.e.* no conditional behavior based on the value of a tensor in the graph, etc.

## L. Implementation Details

All the methods tested in the work share the same actor-critic architecture, for fairness. The reward networks also align in terms of number of parameters, activations, initializations, etc. **Only DiffAIL** adopts a **different reward network** because it is a diffusion model. For DiffAIL (Wang et al., 2023) however, we took the authors' official implementation of the reward learning process, and made modifications to prevent pervasive `NaN` occurrences. Specifically, we replaced Mish activations with LeakyReLU, and added $\epsilon$ padding to the operands of logarithms and denominators. We left the DiffAIL reward architecture untouched otherwise, despite being deeper than those used in other methods. We posited that this would give the diffusion model a fairer chance, despite being misaligned with the rest.

We use the Adam optimizer (Kingma & Ba, 2014) for all experiments.

## M. Main Environments

We report below the dimensionalities of the state and action spaces for the continuous control tasks considered in this work, based on the Gymnasium MuJoCo `v4` environments documentation (Towers et al., 2024).

## N. Ablation Studies

To better understand the design choices underlying our method, we conducted a comprehensive set of ablation studies. These experiments span four random seeds and varying numbers of expert demonstrations, **totaling 92 training runs**. The results, summarized below, provide further empirical support for the architectural and algorithmic decisions presented in the main paper.

We first assess the contribution of the **histogram loss** "Gaussian type" $\ell_{\text{HLG}}$ by replacing it with a Mean Squared Error (MSE) Softmax loss on the `Humanoid` environment—the output embeddings returned by the predictor network $f_\xi$ and prior network $f_\xi^\dagger$ are first wrapped with a softmax, before being compared with the MSE. As shown in FIGURE 5, the MSE Softmax loss fails to produce meaningful learning signals, leading to poor policy performance. In contrast, the histogram loss $\ell_{\text{HLG}}$ loss enables successful and stable training on this challenging high-dimensional benchmark.

Next, we evaluate the generalization capacity of the histogram loss $\ell_{\text{HLG}}$ across environments with *lower* state-action dimensionality the `Humanoid`. As illustrated in FIGURE 6, $\ell_{\text{HLG}}$ replicates the optimal results reported in the main text for non-Humanoid environments, albeit requiring different hyperparameter settings (for $\ell_{\text{HLG}}$) than the `Humanoid`. In particular, the number of bins $N$, support width $a$ and $b$, and Gaussian smoothing factor $\sigma$ must be adapted to the environment. This supports the heuristic that these parameters should **scale with task difficulty**, in a manner similar to entropy target scaling in SAC (Haarnoja et al., 2018). The histogram loss is therefore able to make NGT optimal in more than just the `Humanoid`; we just prioritized the use of pairing losses $\ell$ **without hyper-parameters** unless required. Hence, in the results reported in SECTION 5, we only used $\ell_{\text{HLG}}$ for the `Humanoid`.

We also ablate the **initialization scheme** used for the output embedding layers of the prior and predictor networks. As seen in FIGURE 7, switching from **orthogonal** initialization (Saxe et al., 2013) to Kaiming initialization (He et al., 2015) introduces significant training instability and reduced performance. These results highlight the importance of initialization in preserving gradient flow and inducing stable dynamics in the learned reward model. We refer the reader to SECTION 4.3.1 where we justify the design choice of opting for orthogonal initialization for the output embedding of the prior network $f_\xi^\dagger$.

---

[6] https://pytorch.org/blog/accelerating-pytorch-with-cuda-graphs/

*Table 1.* Default Hyper-parameters for NGT Algorithm

| Hyper-parameter | Default Value |
| --- | --- |
| GPU | True |
| PyTorch's `compile` | False |
| `CudaGraphs` | True |
| Number of parallel environments | 4 |
| Action repeat | 1 |
| Observation normalization | False |
| Number of environment steps | $10^7$ |
| Learning starts at timestep | 0 |
| Evaluation steps | 10 |
| Evaluate every | 10000 |
| Evaluation window buffer size | 20 |
| Clip norm actor | 20.0 |
| Replay buffer size ($|\mathcal{D}|$) | $4 \times 10^6$ |
| Minibatch size ($|\mathcal{B}|$) | 256 |
| Discount factor ($\gamma$) | 0.99 |
| Polyak target smoothing coefficient ($\tau$) | 0.005 |
| Learning rate – policy | $3 \times 10^{-4}$ |
| Learning rate – Q-networks | $1 \times 10^{-3}$ |
| Temperature parameter ($\alpha$) – auto-tune | True |
| Temperature parameter ($\alpha$) – initial value | 0.2 |
| Target entropy ($\mathcal{H}$) | $-|\mathcal{A}|$ (dimension of the action space) |
| Number of gradient steps per update | 1 |
| Number of environment steps per update | 1 |
| Learning rate – reward | $1 \times 10^{-3}$ |
| Spectral normalization | True |
| Gradient penalty | False |
| Output embedding size | 32 |
| Output post-`tanh` rescale | 5.0 |
| $\ell_{\mathrm{HLG}}$ – Support $[a, b]$ | $[-1, 1]$ |
| $\ell_{\mathrm{HLG}}$ – Number of bins $N$ | 21 |
| $\ell_{\mathrm{HLG}}$ – agent-side $\sigma$ | 0.05 |
| $\ell_{\mathrm{HLG}}$ – expert-side $\sigma$ | 0.25 |
| Number of behavioral cloning iterations | $10^7$ |

Another critical component is **spectral normalization**. We remove spectral normalization from the prior and predictor networks and observe a complete failure of training, as reported in FIGURE 8. This indicates that constraining the Lipschitz constant of these networks is essential for stable and reliable reward learning. Note, **gradient penalization was not required**, unlike for DAC/SAM (Blondé et al., 2020).

Finally, we study the impact of the output embedding dimensionality in the reward model by varying it across {8, 16, 32, 64}. As shown in FIGURE 9, performance is relatively stable for sizes 16 and above, but **degrades substantially** for 8-dimensional embeddings. This suggests that excessively compressing the output representation harms expressiveness and impedes optimization.

Taken together, these ablations substantiate the design principles of our approach and further validate the empirical findings presented in the main paper.

*Table 2.* Dimensionalities of the state and action spaces for selected Gymnasium environments. These environments are commonly used benchmarks for continuous control in reinforcement learning.

| Environment | State Dim. | Action Dim. |
|---|---|---|
| Ant-v4 | 111 | 8 |
| HalfCheetah-v4 | 17 | 6 |
| Pusher-v4 | 23 | 7 |
| Walker2d-v4 | 17 | 6 |
| Humanoid-v4 | 376 | 17 |

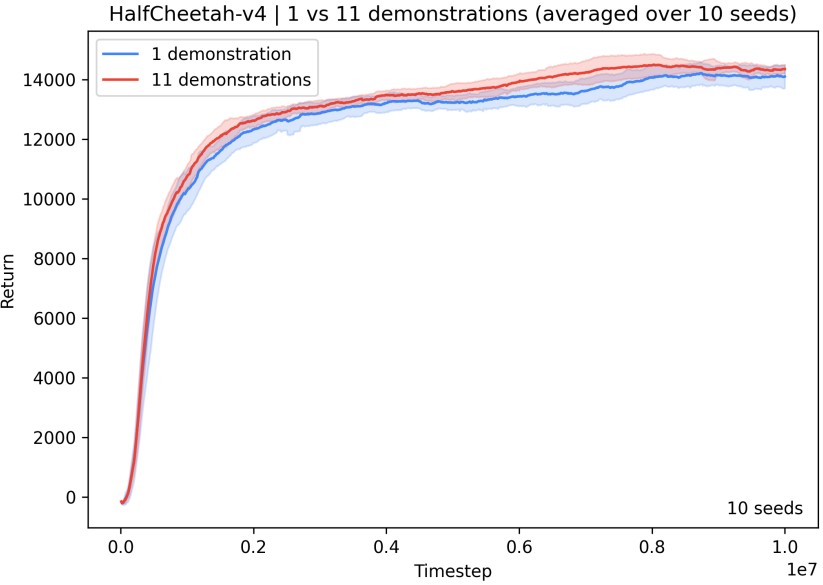

*Figure 4.* Performance of NGT on `HalfCheetah-v4` with 1 and 11 demonstrations, averaged over 10 seeds. Shaded regions show standard deviation across seeds.

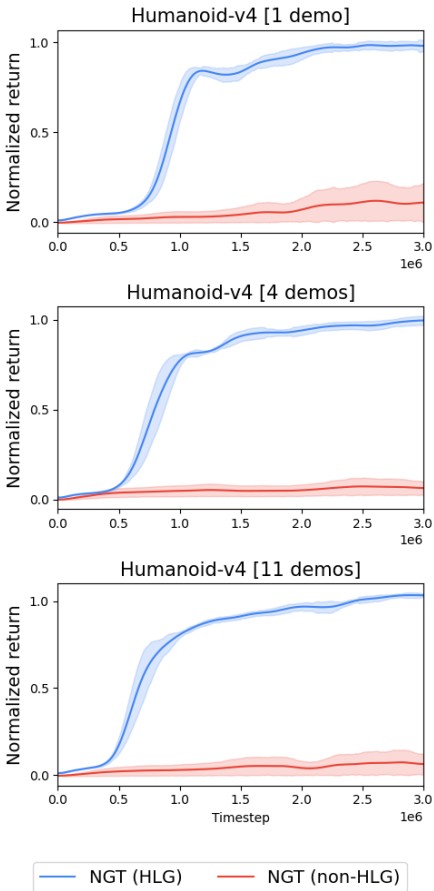

*Figure 5.* Comparison of the histogram loss $\ell_{\mathrm{HLG}}$ and a Mean Squared Error (MSE) Softmax loss in NGT on the `Humanoid` environment. The MSE variant fails to yield meaningful learning, while $\ell_{\mathrm{HLG}}$ enables successful and stable training.

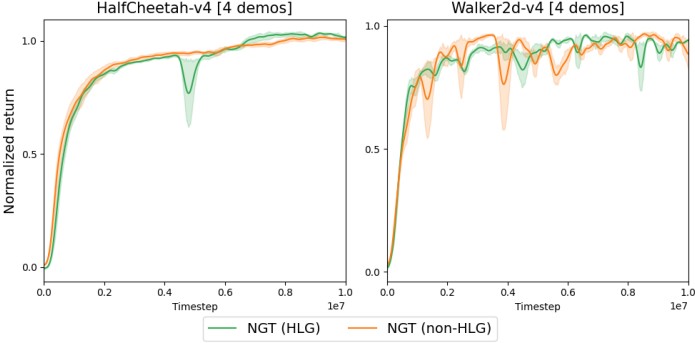

*Figure 6.* Performance of the histogram loss $\ell_{\mathrm{HLG}}$ on non-`Humanoid` environments. Optimal results are recovered when adapting hyper-parameters such as support width, number of bins, and smoothing factor $\sigma$, highlighting the need for environment-specific scaling.

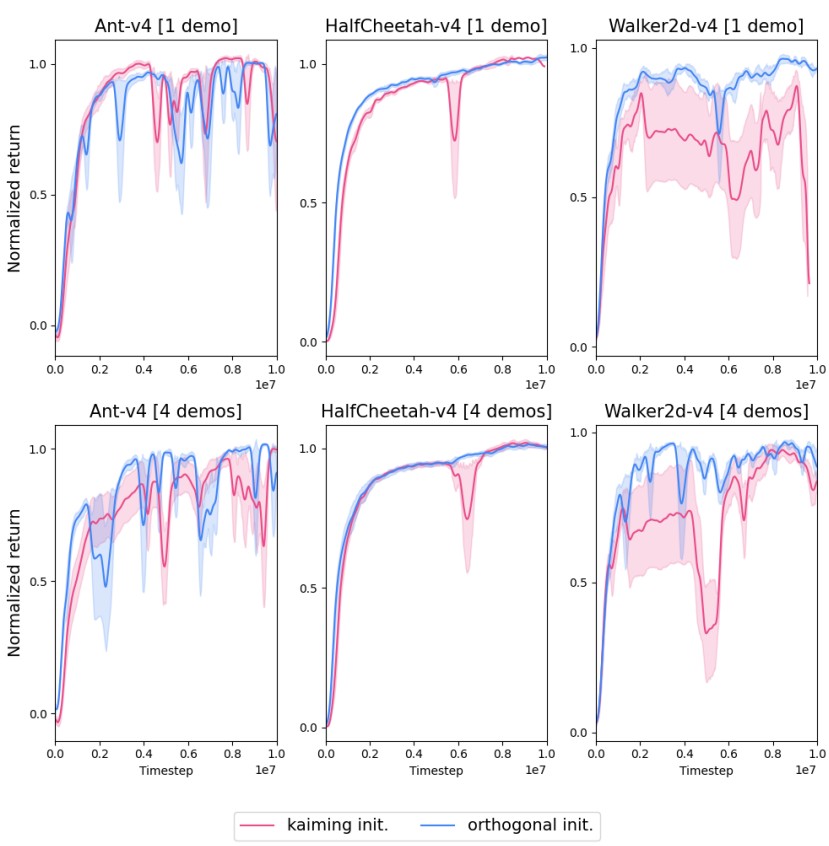

*Figure 7.* Impact of output embedding initialization scheme. Orthogonal initialization leads to higher stability and performance, whereas Kaiming initialization results in degraded learning and increased instability.

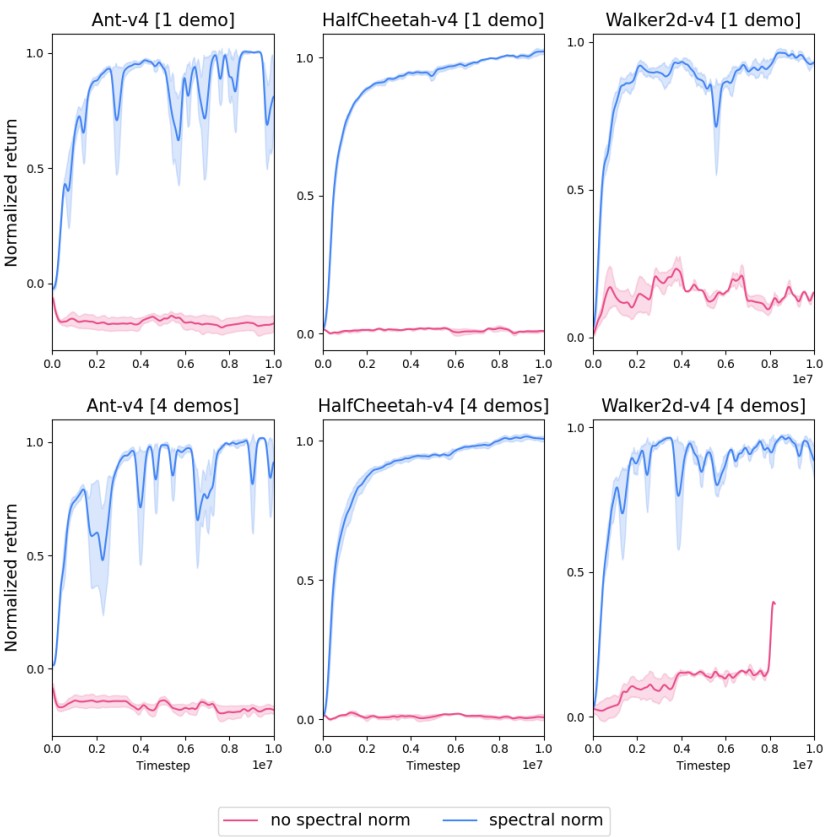

*Figure 8.* Effect of removing spectral normalization from the reward model's prior and predictor networks. Training becomes unstable and collapses completely, underscoring the necessity of spectral normalization for stable optimization.

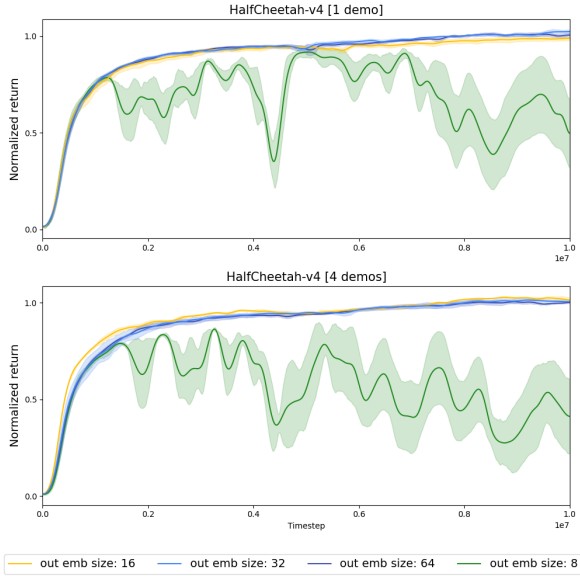

*Figure 9.* Effect of output embedding dimensionality in the reward model. Performance is robust for sizes 16, 32, and 64, but degrades significantly at dimension 8.

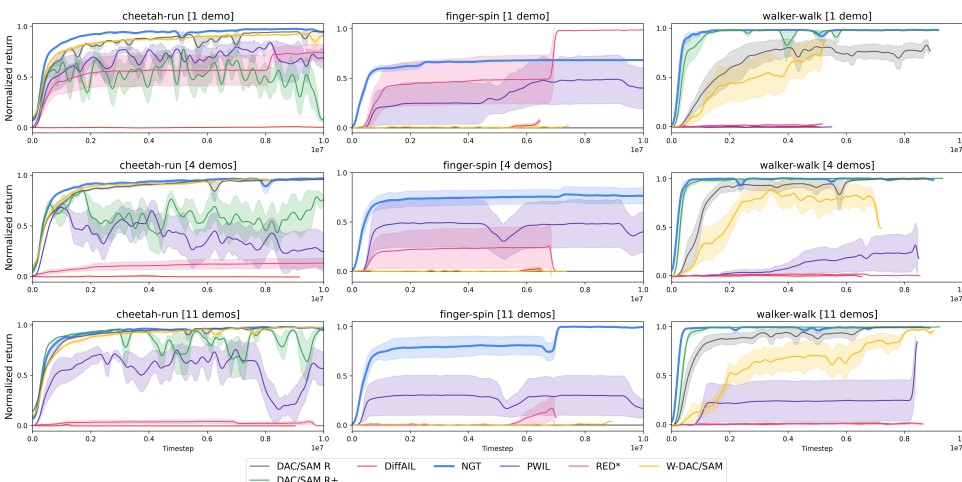

*Figure 10.* Performance of NGT and baseline methods on three DeepMind Control Suite tasks over various environments and numbers of demonstrations. DMC environments introduce greater variability in the initial state distribution, providing a more stringent test of generalization. NGT achieves strong and stable returns across all tasks, demonstrating robustness to increased stochasticity and variation in starting conditions.

## O. Extra Environments

To further evaluate the robustness and generalization capacity of imitation learning agents, we extend our experiments to three environments from the DeepMind Control Suite (DMC) (Tassa et al., 2018): `walker-walk`, `cheetah-run`, and `finger-spin`. Compared to Gymnasium tasks, DMC environments exhibit **greater stochasticity in their initial state distributions**. This increased variability poses a significantly **harder generalization challenge**, as agents must adapt to a broader range of starting conditions rather than overfitting to narrow behavioral modes. This makes DMC a natural and meaningful extension for benchmarking IL methods, particularly in the **low-data regime**. Agents must not only mimic expert behavior but generalize it across unseen trajectories and perturbed states—highlighting the inductive biases and stability of the learning algorithm.

FIGURE 10 shows performance curves for NGT and several baseline methods across these three DMC tasks. We observe that NGT consistently achieves strong returns and exhibits remarkable training stability. In contrast, baseline methods often show high variance or fail to approach expert behavior at all, failing to generalize effectively under the broader initial state distributions. These results **further validate the generalization ability and sample efficiency of NGT** in more challenging, high-variance control settings.

# P. Speeds

We report the computational speed of the imitation learning methods we compare in SECTION 5, measured in **steps per second** (sps) at 200k environment time-steps. This allows us to assess their efficiency on two high-dimensional continuous control tasks: `Humanoid-v4` and `Walker2d-v4`.

Since the reported speeds are expressed in steps per second, **higher is better**.

*Table 3.* Speed (steps per second) of compared methods at 200k time-steps.

| Method | Humanoid | Walker2d |
|---|---|---|
| PWIL | 631 | 749 |
| DiffAIL | 659 | 929 |
| DAC/SAM | 741 | 953 |
| MMD-DAC/SAM | 749 | 977 |
| W-DAC/SAM | 763 | 978 |
| **NGT** | **712** | **967** |

`RED*` exhibits runtime performance on par with NGT.

NGT demonstrates competitive runtime efficiency with near-horizontal speed curves by 200k time-steps in both environments. This suggests that its performance stabilizes early, avoiding the degradation seen in other methods. DiffAIL, by contrast, exhibits a noticeable drop in speed over time, indicative of its steeper negative slope and greater runtime overhead. `DAC/SAM` achieves high speeds overall, though its reliance on gradient penalization introduces an initial computational burden. Nevertheless, by 200k steps, its speed curve also flattens, similar to NGT. PWIL is significantly slower, reflecting its less efficient reward computation. Among all methods, `MMD-DAC/SAM` and `W-DAC/SAM` are the fastest by a small margin, particularly in the more demanding `Humanoid-v4` task.

Overall, the results show that **NGT maintains a strong balance between computational efficiency and learning performance**, with speed profiles comparable to the fastest baseline methods while avoiding the pitfalls of runtime degradation.

