# OpenReview forum: "Noise-Guided Transport: Imitation Learning from Random Priors"
_ICML.cc/2026/Conference — ICML 2026 regular_

### Official Review · Reviewer_wDca · 2026-03-05

**Soundness:** 2
**Presentation:** 2
**Significance:** 3
**Originality:** 2
**Overall Recommendation:** 4
**Confidence:** 3

**Summary:**

This paper proposes the Noise-Guided Transport (NGT) that is useful for imitation learning with limited data by training a reward model to distinguish in between expert trajectory and generated ones. This paper derives an adversarial training objective from learning from random priors to modify the reward learning objective, and theoretically derives bounds depending on $\sigma$. Empirical results demonstrate that NGT is effective on high dimensional control tasks with limited expert trajectories.

**Compliance With Llm Reviewing Policy:**

Affirmed.

**Final Justification:**

I will keep my score.

**Key Questions For Authors:**

- Some of the imitation learning methods such as GAIL (which also have a adversarial training framework like NGT) may suffers from unstable training issue. What is the stability guarantee and the rate of convergence of NGT?
- In line 413 the paper refers to NGT's performance over 10 runs in figure 4, which is kind of ambiguous since figure 4 is in the appendix far later and is not on the Appendix O of the previous sentence, so the authors may consider rehearsing the referring wording.
- There is a typo in line 354 with [0..19]
- Why in figure 2, the training curves for BC are horizontal lines? Does that mean BC gets into a local minimum?

**Limitations:**

Yes

**Strengths And Weaknesses:**

## Strengths
- This method combines optimal transport theory into imitation learning to improve the sample efficiency
- The authors derive clear bound on the hyper parameters and give instructions on how to decide their values
- Numerical experiments validate the effectiveness of NGT comparing to other imitation learning methods


## Weaknesses
- The paper may have a separate paragraph clearly state the difference and contribution of this method comparing with GAIL. GAIL also have a discriminator (like the reward function) for matching the expert and agent. May also need theoretical comparison with GAIL.
(The related work mentioned GAIL-DAC which is also compatible with limited data in practice. So what's the contribution of NGT comparing to DAC?)

- The results of GAIL-DAC in their original paper are also presented with 4 demonstrations. However in Figure 2 the results for GAIL-DAC seems much lower than the results in GAIL-DAC paper in 4 demos, which raise the question of the fairness and soundness of the experiments

---

> ### Author Rebuttal · Authors · 2026-03-31
>
> We thank the reviewer for the feedback and questions.
> - **W1: Difference from GAIL/DAC.** NGT is not a reparameterization of GAIL/DAC. GAIL and DAC learn rewards through a GAN-style binary discriminator and optimize a JS-based adversarial objective. In contrast, NGT starts from predictor-prior discrepancy under learning from random priors, and Section 4.2 shows that, under the 1-Lipschitz restriction, the resulting reward-learning objective coincides with an EMD objective. The difference is therefore both methodological and theoretical. We agree, however, that this contrast could be stated more directly in the paper, and we will revise the presentation accordingly.
> - **W2: About the DAC baseline.** All methods were implemented on the same SAC-based actor-critic backbone and evaluated under the same training protocol. This was done deliberately to isolate the effect of the reward-learning method under a common control stack. The curves are therefore not directly comparable to those reported in the original DAC or SAM papers, since those works use different base algorithms (DAC uses TD3 and SAM uses DDPG, whereas we use SAC throughout). We chose SAC because it is a strong and widely used off-policy baseline, and because it supports automatic entropy tuning for exploration. Within this unified setup, we aimed for the fairest reproduction possible across methods.
> - **Q1: Stability guarantee and convergence rate.** Our theoretical guarantees concern the reward-learning objective rather than the full coupled off-policy actor-critic dynamics. Under the 1-Lipschitz restriction, Section 4.2 connects the objective to an EMD objective and establishes concentration of its empirical estimate around its expectation. We do not claim a global convergence-rate theorem for the full deep off-policy training procedure. At a practical level, one motivation for NGT was precisely the instability of adversarial reward learning in the off-policy regime: unlike GAN-style approaches, which often rely on gradient-penalized discriminator training to maintain regularity, NGT is constructed to promote reward regularity through the predictor-prior construction derived in the paper, which we found empirically more stable.
> - **Q2: Reference to Figure 4.** We thank the reviewer for pointing this out. We will revise the wording so that the reference points more directly and unambiguously to the appendix figure reporting the 10-seed experiment.
> - **Q3: Notation in line 354.** We thank the reviewer for pointing this out. We will replace `[0..19]` with the more standard notation `{0, ..., 19}`.
> - **Q4: Why is BC horizontal in Figure 2?** BC is trained offline on a fixed dataset of demonstrations. Unlike the other methods, it is not updated through online interaction with the environment. The flat curves therefore reflect the protocol rather than convergence to a local minimum. The y-axis shows the final performance of the BC policy after offline training.
> We thank the reviewer again for the questions and suggestions, which will help improve the clarity of the final version.

---

> > ### Author Rebuttal · Reviewer_wDca · 2026-04-01
> >
> > Thank you for clarifying my concerns. I have raised my score.

---

### Official Review · Reviewer_rSj4 · 2026-03-11

**Soundness:** 4
**Presentation:** 4
**Significance:** 3
**Originality:** 3
**Overall Recommendation:** 5
**Confidence:** 4

**Summary:**

This paper introduces Noise Guided Transport, a reward learning framework that enables robust policy learning with low amounts of expert data. The method is based on 'learning from random priors', a technique that uses a random frozen network, and its trainable equivalent, using a loss or discrpancy metric between them as a proxy for density estimation. They provide theoretical guarantees and reasoning on how to design the associated loss, and finish by testing their method on gymnasium continuous control tasks.

**Compliance With Llm Reviewing Policy:**

Affirmed.

**Final Justification:**

The paper is clear, interesting and easy to follow while bringing a real contribution. I maintain my accept.

**Key Questions For Authors:**

1. The paper presents the method as lightweight and stable, however the $l_{HLG}$ loss adds 4 hyperparameters. How sensitive is the method to these hyperparameters? Have you found any performance gains or instabilities coming from the design of your distributional loss?
2. Why do you think 'simple' losses fail on humanoid?
3. Have you made any attempts to keep scaling up to higher DoF tasks? Since there seems to be a breaking point after which $l_{HLG}$ becomes mandatory, it would be interesting to see if it in turn reaches a ceiling at some point.

**Limitations:**

yes

**Strengths And Weaknesses:**

## Strengths
- First, the paper is extremely well structured, easy to read and to follow. The setting, background and method are well explained and justified, the design choice come from sound theoretical groundings and all necessary ablations are provided in the appendix.
- The method is interesting, and the use of the random prior seems to be able to alleviate major issues with the basic formulation of imitation learning. It is also not something I have observed in control papers, bridging the gap between general AI techniques and their application to control tasks is important.
- Generalization in continous control is a very valuable property: even if nowadays VLAs seem to have somehow the capacity to do few / zero shot learning, having a principled way to do this from scratch could be useful in many applications.
## Weaknesses
- The necessity for the distributional loss adds a layer of complexity to the tuning, which can already be high in adversarial setting.
- The evaluation of the methods is only done on 'toy' environments. Since the authors were able to scale up to humanoid v4 which is 17 DoF, testing on robotics environments like maniskill, or even scaling up to Adroit (~30 DoF) would have made an even stronger case for the method by showing performance on direct applications.

Overall, the paper seems impactful, is a very easy read and shows clear cut results, and even if more applicative tests would have been nice, it does not take anything away from the quality of the work.

---

> ### Author Rebuttal · Authors · 2026-03-31
>
> We thank the reviewer for the very positive assessment and thoughtful questions.
> - **Q1: sensitivity to the distributional loss hyperparameters.** The HL-Gaussian distributional loss introduces additional hyperparameters: the support range $[a, b]$, the number of histogram bins, and the Gaussian standard deviations controlling the amount of label smoothing in the target distributions. We also allow different standard deviations on the expert and agent sides, which makes it possible to use either symmetric or one-sided label smoothing. Empirically, we did not observe meaningful sensitivity to the support range or the number of bins once they were set to reasonable values. The main sensitivity was to the Gaussian standard deviations. When they are too large, the supervision becomes too coarse; when they are too small, the benefit over a sharper loss largely vanishes. In the harder environments, this smoothing was useful because it made the reward-learning signal denser and less brittle. We will clarify in the revision that this is the main practical tradeoff introduced by the distributional loss.
> - **Q2: why do simpler losses fail on Humanoid?** Our interpretation is that the key issue is the difficulty of obtaining a useful learning signal early in training. In Humanoid, most behaviors encountered during training are far from expert-like and often lead to early failure, so the learned reward can become dominated by uniformly poor outcomes. In that regime, a sharper loss provides little structure beyond penalizing failure. The distributional loss helps because its smoothing makes the reward-learning problem less binary and provides a richer gradient signal, which appears to matter much more in Humanoid than in the easier environments.
> - **Q3: scaling to higher-DoF settings.** We have not yet attempted to scale the method further to substantially higher-DoF settings, but this is a direction we are very interested in. One relevant extension, in line with the reviewer’s suggestion, is goal-conditioned robotics. In the low-demonstration regime, we expect such settings to require more explicit goal handling, since simply including the goal in the state may not be sufficient unless the expert and the learner are always conditioned on exactly the same goal. Another extension we find especially compelling is musculo-skeletal control (with the `myosuite` library, containing the environments introduced in the NeurIPS “MyoChallenge” competitions). These systems do not always involve more DoFs, but they introduce a different and significant challenge: the action space is over-actuated, with many muscles contributing to each joint and individual muscles affecting multiple joints. This setting also fits the low-demonstration regime well, since obtaining real demonstrations at scale is difficult in musculo-skeletal domains and requires specialized kinesiology setups and expertise.
> We thank the reviewer again for the very positive assessment and for the depth of engagement with the technical aspects of the paper.

---

> > ### Author Rebuttal · Reviewer_rSj4 · 2026-04-02
> >
> > I thank the reviewers for their answers to my questions and the interesting added information. I will maintain my score which is already an accept!

---

### Official Review · Reviewer_GZq5 · 2026-03-13

**Soundness:** 3
**Presentation:** 3
**Significance:** 3
**Originality:** 3
**Overall Recommendation:** 4
**Confidence:** 3

**Summary:**

The paper proposes a method to do imitation learning with limited demonstration based on casting it as optimal transport and using adversarial training.

**Compliance With Llm Reviewing Policy:**

Affirmed.

**Key Questions For Authors:**

ROT and MAAD are both imitation learning based optimal transport methods, how does the method compare?

How far does the learned potential actually deviate from 1-Lipschitz during training, and does this deviation correlate with performance degradation?

How sensitive is performance to hyperparameters and is there a principled selection criterion that would work across environments?

**Limitations:**

yes.

**Strengths And Weaknesses:**

Soundness
The paper is technically sound. The claims are well supported. The experimental design is good with a good number of settings. The comparison does not include some recent strong baselines in low-data IL, such as ROT or MAAD.

Presentation
The presentation is good. The progression from the problem of learning from random priors, through the adversarial objective, to the OT equivalence, and finally to practical implementation is easy to follow.

Significance
The problem setting of imitation learning with a small number of demonstrations is practical. The theoretical analysis is significant. The experiments are only limited to MuJoCo style continuous control and the gains over the best baselines are not always dramatic outside of Humanoid.

Originality
The work is original in both the method and analysis.

---

> ### Author Rebuttal · Authors · 2026-03-31
>
> We thank the reviewer for the positive assessment and for the thoughtful questions. We address them below.
> - **Q1: comparison to ROT/MAAD.** ROT and MAAD are indeed relevant points of comparison. A key difference, however, is that both are on-policy methods, whereas NGT is online and off-policy, so a direct comparison would not be fully apples-to-apples in terms of data usage and interaction regime. There is also a methodological difference: ROT and MAAD compute optimal transport through Sinkhorn-style primal OT machinery, whereas NGT learns a potential and is therefore closer to a dual OT formulation. In addition, MAAD requires learning an inverse dynamics model, which introduces an additional source of approximation error and tuning complexity. Adapting such methods to the off-policy setting would be an interesting direction for future comparison, and we will clarify this distinction in the paper.
> - **Q2: how far does the learned potential deviate from 1-Lipschitz during training (Part 1), and does this correlate with performance degradation (Part 2)?** Part 1: To probe how far the learned potential departs from the 1-Lipschitz regime during training, we ran diagnostic experiments in three environments, Walker2d, Ant, and HumanoidStandup, using a one-sided gradient penalty on top of spectral normalization. Since this regularizer is zero unless the norm of the gradient of the reward network with respect to its inputs exceeds 1, it provides an indicator of violations of the 1-Lipschitz target under this test. All three environments yielded the same outcome: the diagnostic remained zero at all logged evaluation points, suggesting no detectable violation under this measure in those runs. Because the value stayed at zero, it did not affect optimization in those experiments and served only to monitor the learned potential. We can report this empirical result more explicitly in the revision. Part 2: Under the diagnostic described above, we did not observe detectable violations when spectral normalization was enabled: the one-sided gradient-penalty indicator remained zero at all logged evaluation points in Walker2d, Ant, and HumanoidStandup. By contrast, when spectral normalization is removed, there is no mechanism enforcing the Lipschitz condition, and the appendix ablation shows a substantial drop in performance. This therefore covers the two extreme cases we evaluated empirically: a regime in which violations are not detected under our diagnostic and performance remains strong, and a regime in which Lipschitz control is removed and performance degrades markedly. What we have not fully mapped out in the current paper is the intermediate regime between these two extremes. Doing so would require a systematic sweep over the strength of the Lipschitz-control mechanism itself. In that respect, these observations are consistent with prior evidence from Blondé et al. (2022) “Lipschitzness Is All You Need”, who report that stronger Lipschitz control correlates with improved imitation-learning performance. We therefore view our results as consistent with that broader picture, while being careful not to overclaim beyond the experiments included in the paper.
> - **Q3: sensitivity to hyperparameters and principled selection across environments**. The main method-specific hyper-parameters in NGT are the choice of pairing loss $\ell$ and the output embedding dimension. By contrast, spectral normalization and orthogonal initialization are intended as fixed design choices rather than tuning axes; we motivate them in the main text and support them empirically in the appendix. For the embedding dimension, our ablations suggest limited sensitivity once very small values are avoided: performance degrades at low dimension (e.g., 8), while values in the 32-64 range appear broadly reliable across the studied settings. For the pairing loss, we view the Huber loss as the default choice because of its robustness and favorable regularity properties. Histogram losses can provide additional robustness in more difficult settings such as Humanoid, but they introduce extra hyperparameters, so we view them as a targeted extension rather than the default configuration. We will revise the paper to make this distinction clearer.
> We thank the reviewer again for the positive assessment and the thoughtful feedback.

---

> > ### Author Rebuttal · Reviewer_GZq5 · 2026-04-04
> >
> > I appreciate the author's response to my questions. The responses answer the question. I will maintain my positive evaluation.

---

### Official Review · Reviewer_Q3uv · 2026-03-20

**Soundness:** 3
**Presentation:** 4
**Significance:** 3
**Originality:** 3
**Overall Recommendation:** 5
**Confidence:** 5

**Summary:**

The paper studies low-data imitation learning for continuous control, where only a small number of expert demonstrations are available and standard behavioral cloning often fails to generalize. It proposes Noise-Guided Transport (NGT), an off-policy method that learns a reward from the discrepancy between a trainable predictor and a frozen random prior, using both expert and agent data. The main technical contribution is to show that this reward-learning objective can be interpreted through optimal transport, with the learned potential tied to an earth mover’s distance style objective under a Lipschitz constraint. Empirically, the paper reports strong performance across standard control benchmarks, including difficult Humanoid settings and cases where only states or transitions are available, while remaining relatively lightweight and avoiding gradient-penalty regularization.

**Compliance With Llm Reviewing Policy:**

Affirmed.

**Final Justification:**

Most of my comments and the associated replies relate to the presentation aspect. As such I increased my presentation score. I keep my overal recommendation intact (Accept).

**Key Questions For Authors:**

1) Can you clarify the exact novelty claim relative to Random Expert Distillation (RED)? The paper clearly builds on the random-prior line of work and even defines RED* by using only the expert-side term of the NGT objective, so it would help to isolate what is fundamentally new beyond the addition of agent-side updates and the optimal transport (OT) interpretation.

2) How important is the optimal transport (OT) interpretation in practice, given that the paper does not enforce a perfect 1-Lipschitz condition? The theory is framed around the 1-Lipschitz potential, but the practical implementation only aims to keep the potential “close enough to 1” via spectral normalization. If the authors can better justify this gap empirically or theoretically.

3) How robust is NGT to the practical design choices that seem essential in the appendix? The ablations suggest that spectral normalization is necessary, orthogonal initialization matters, low embedding dimension hurts, and the histogram loss requires environment-specific tuning. Please show that the method remains stable under a broader range of reasonable settings.

4) How fair are the baseline comparisons, especially for RED and DiffAIL? The paper uses an adapted RED* reward scheme instead of the original RED temperature tuning, and notes that DiffAIL lacked a Humanoid configuration so the setup was adapted from Ant. A clearer discussion of how much these implementation choices affect the comparison could help support the empirical claims.

**Limitations:**

The authors discuss several technical limitations, but I would encourage a clearer discussion of broader scope limitations and any potential societal risks if the method were applied beyond simulated control settings.

**Strengths And Weaknesses:**

Soundness: The paper is technically solid overall. The method is clearly specified, the paper gives a formal optimal transport (OT) interpretation under a 1-Lipschitz constraint, provides concentration-style guarantees, and supports the claims with broad empirical comparisons and ablations. A weakness, however, is that the strongest theoretical story relies on the idealized 1-Lipschitz setting, while in practice the authors explicitly say they only keep the potential “close enough” to 1, so the theory-to-practice connection is not fully tight.

Presentation: The paper is generally well structured and easy to follow, with a clean progression from background on random priors, to the reward objective, to the OT interpretation, and then to experiments. The algorithm box and appendices also help make the implementation more clear. The main weakness is that some practically crucial details are deferred or understated relative to the main narrative, especially the dependence on spectral normalization, initialization, and the histogram loss for Humanoid, which makes the “simple and easy to tune” message feel somewhat stronger than the full paper supports.

Significance: The paper addresses an important problem about imitation learning with very limited expert data and the reported results on low-data continuous control, including state-only settings and Humanoid, make the work potentially useful for settings where demonstrations are scarce. At the same time, the demonstrated impact is still somewhat bounded by the benchmark setting. The experiments are all in simulated continuous-control domains with SAC-generated experts, so the broader practical significance beyond this regime is promising but not yet fully established.

Originality: The paper’s originality is real but should be framed carefully. The strongest contribution is the synthesis of a random-prior reward model with a two-sided expert-versus-agent objective and an OT-based interpretation, which is a meaningful conceptual combination. The main weakness is that the core random frozen prior mechanism is not new by itself, since the paper explicitly positions NGT after Random Network Distillation (RND) and Random Expert Distillation (RED), so the novelty is more incremental and compositional than foundational.

---

> ### Author Rebuttal · Authors · 2026-03-31
>
> We thank the reviewer for the thoughtful and positive assessment.
> - **On the role of practical details and the paper’s presentation.** The reviewer notes that some practically important details are deferred relative to the main narrative. This reflects our intended structure: we first introduce the method, then present the theoretical results under the relevant conditions, and finally explain the practical design choices used to approximate those conditions. Because off-policy adversarial imitation learning is highly sensitive to implementation details, we felt these choices should be discussed explicitly in the main text. That said, we agree their importance could be signaled earlier, and we will revise the introduction accordingly.
> - **On the “simple and easy to tune” claim.** The reviewer raises a fair point. Our intent was to distinguish fixed design choices from tunable hyper-parameters. Spectral normalization and orthogonal initialization are part of the method and are not meant to be tuned across instantiations. By contrast, the histogram loss used on Humanoid corresponds to a high-dimensional special case and may require task-specific tuning. We will revise the wording to make this distinction clearer and avoid overstating ease of tuning in such cases.
> - **On novelty and positioning within the random-prior lineage.** We believe we have cited the works from that lineage appropriately, and have positioned the objectives optimized by the most relevant algorithms from that lineage (RND and RED) with respect to NGT. We agree with the reviewer that NGT should be positioned as part of the random-prior lineage rather than as a foundational departure from it. We will revise the paper to make this positioning more explicit.
> - **Q1: position writh respect to RED.** As the reviewer notes,  NGT optimizes a two-sided expert-versus-agent objective. If the agent-side term is removed, the remaining objective matches the one-sided expert objective optimized by RED in the _offline_ setting. Since our setting is online and off-policy, the variant we denote “RED*” corresponds to optimizing that one-sided objective _online_. The key step beyond RED is therefore not the frozen-prior mechanism itself, but the addition of the agent-side term, which makes reward learning adversarial, together with the OT interpretation of that objective. We will clarify this point  in the revision.
> - **Q2: theory-to-practice gap.** To assess this gap empirically (i.e., whether the learned reward remains close to 1-Lipschitz in practice), we ran diagnostic experiments in three environments using a one-sided gradient penalty on top of spectral normalization. This regularizer is zero unless the norm of the the gradient of the reward with respect to its input exceeds 1, so it provides an indicator of violations of the 1-Lipschitz target under this test. All three environments yielded the same outcome: the diagnostic remained zero at all logged evaluation points, suggesting no detectable violation under this measure in those runs. Since the diagnostic stayed at zero, it did not contribute gradients in those experiments and served only as a diagnostic. We agree that this does not fully close the theory-to-practice gap, and we will make that limitation more explicit in the revision. At the same time, these results suggest that the constraints used in NGT are sufficient to keep training in the intended regime.
> - **Q3: robustness to the practical design choices.** As the reviewer notes, the main practical design choices are supported by ablations in the appendix. Our goal there was to test the components most directly tied to the theory-to-practice gap and to empirical stability. Among these, spectral normalization is the most critical, as it is the main mechanism used to keep the reward network close to the Lipschitz regime required by the theory. Orthogonal initialization and sufficient embedding dimension also contribute to robustness, while the histogram loss is only introduced for the particularly challenging Humanoid setting. We will revise the paper to make clearer which choices are method-defining, and which primarily affect performance margins rather than overall stability.
> - **Q4: RED and DiffAIL.** We agree that the notation RED* may be misleading. A clearer presentation would be to treat this result as an ablation corresponding to setting the ascent weight in the NGT objective to zero, rather than as a separate baseline. We are happy to revise the presentation accordingly. For DiffAIL on Humanoid, the original paper does not report results on that environment, so we used the Ant hyper-parameters, since Ant is the highest-dimensional environment considered by the authors. We agree that this adaptation should be stated more explicitly, and we will clarify it in the revision.
> - **Scope, limitations, societal risks.** We agree and will strengthen this discussion in the revision.

---

> > ### Author Rebuttal · Reviewer_Q3uv · 2026-04-03
> >
> > Thanks for your response. I adusted my score.

---

### Decision · Program_Chairs · 2026-04-30

**Decision:**

Accept (regular)

**Comment:**

This paper is technically sound, well-written, reasonably novel, and useful to at least some fraction of the ICML community.